# Resveratrol: A Review on the Biological Activity and Applications

**Ludovic Everard Bejenaru [1], Andrei Biță [1], Ionela Belu [2,\*], Adina-Elena Segneanu [3], Antonia Radu [4], Andrei Dumitru [5], Maria Viorica Ciocîlteu [6], George Dan Mogoşanu [1] and Cornelia Bejenaru [4]**

[1] Department of Pharmacognosy & Phytotherapy, Faculty of Pharmacy, University of Medicine and Pharmacy of Craiova, 2 Petru Rareş Street, 200349 Craiova, Romania; ludovic.bejenaru@umfcv.ro (L.E.B.); andreibita@gmail.com (A.B.); george.mogosanu@umfcv.ro (G.D.M.)

[2] Department of Pharmaceutical Technology, Faculty of Pharmacy, University of Medicine and Pharmacy of Craiova, 2 Petru Rareş Street, 200349 Craiova, Romania

[3] Institute for Advanced Environmental Research, West University of Timişoara (ICAM–WUT), 4 Oituz Street, 300086 Timişoara, Romania; adina.segneanu@e-uvt.ro

[4] Department of Pharmaceutical Botany, Faculty of Pharmacy, University of Medicine and Pharmacy of Craiova, 2 Petru Rareş Street, 200349 Craiova, Romania; antonia.radu@umfcv.ro (A.R.); cornelia.bejenaru@umfcv.ro (C.B.)

[5] Department of Medical Assistance and Kinetotherapy, Faculty of Science, Physical Education and Informatics, University Center of Piteşti, National University for Science and Technology Politehnica Bucharest, 1 Târgu din Vale Street, 110040 Piteşti, Romania; andrei.dumitru@upit.ro

[6] Department of Analytical Chemistry, Faculty of Pharmacy, University of Medicine and Pharmacy of Craiova, 2 Petru Rareş Street, 200349 Craiova, Romania; maria.ciocilteu@umfcv.ro

\* Correspondence: ionela.belu@umfcv.ro; Tel.: +40-351-443-507

**Abstract:** Resveratrol (RSV), a naturally occurring phytoalexin, is the most important stilbenoid synthesized by plants as a defense mechanism in response to microbial aggression, toxins, or ultraviolet radiation. RSV came to the attention of researchers both as a potential chemopreventive agent and a possible explanation for the low incidence of cardiovascular disease (CVD) in French people with a high-fat diet. RSV is mainly administered as a food supplement, and its properties are evaluated in vitro or in vivo on various experimental models. RSV modulates signaling pathways that limit the spread of tumor cells, protects nerve cells from damage, is useful in the prevention of diabetes, and generally acts as an anti-aging natural compound. It was highlighted that RSV could ameliorate the consequences of an unhealthy lifestyle caused by an exaggerated caloric intake. This paper reviews the evidence supporting the beneficial effect of RSV for various pathological conditions, e.g., neoplastic diseases, neurodegeneration, metabolic syndrome, diabetes, obesity, CVDs, immune diseases, bacterial, viral, and fungal infections. The study also focused on the chromatographic analysis of *trans*-RSV (*t*RSV) in Romanian wine samples, providing a comprehensive overview of *t*RSV content across different types of wine.

**Keywords:** resveratrol; stilbenoids; grapes; wine; biological activity; applications

## 1. Introduction

Stilbenes are natural secondary metabolites, also known as phytoalexins. Stilbenoids, the hydroxylated derivatives of stilbenes, have in their structure two benzene nuclei connected by an ethylene bridge, which belong to the category of polyphenolic compounds that are biosynthesized in the same way as flavonoids. Due to their role as phytoalexins, stilbenoids are very important in the defense of plants against pathogens. Up to now, more than 1000 stilbenoids have been isolated and identified, of which the best known are the natural components resveratrol (RSV, 3,5,4'-trihydroxy-stilbene), piceid, piceatannol, pterostilbene, astringin, viniferin, pallidol, hopeaphenol. Stilbenoids, primarily derived

from grapes, are compounds of significant pharmacological interest because they offer numerous health benefits, particularly in preventing and potentially treating various chronic diseases related to aging, through antitumor, antioxidant, anti-inflammatory, cardioprotective, neuroprotective, antidiabetic action [1–3].

The two natural isomers of RSV, *cis*-RSV (*c*RSV) and *trans*-RSV (*t*RSV), are biosynthesized almost entirely in grape skins; the highest concentrations of RSV were determined before the ripening of the grapes. *t*RSV has the highest biological activity, with *c*RSV occasionally being identified only in some wines. It is believed that *c*RSV results from an enzymatic reaction during the wine fermentation process or by exposure of *t*RSV to ultraviolet (UV) radiation. *t*RSV is relatively stable and is preferred for chemical analyzes and syntheses [1].

RSV came to the attention of researchers approximately five decades ago, and it is both a potential chemopreventive agent and a possible explanation for the so-called "French paradox"—the low incidence of cardiovascular disease (CVD) among the French, a people with a high-fat diet as a consequence of moderate consumption of red wine (RW) during meals—and its cardioprotective properties are well-known [1,4].

RSV is mainly administered as a food supplement, and its properties are evaluated in vitro or in vivo on various experimental models. RSV has potential beneficial effects in various diseases, e.g., neoplastic diseases; Alzheimer's disease (AD); Parkinson's disease (PD); diabetes, obesity; CVDs (hypertension); osteoporosis; and bacterial, viral, and fungal infections (Figure 1) [5–7]. In particular, RSV has been found to affect the expression of several genes, including genes encoding cytokines, caspases, matrix metalloproteinases (MMPs), adhesion molecules, and growth factors. Moreover, RSV can modulate the activity of several signaling pathways, such as phosphatidylinositol 3'-kinase (PI3K)/Akt, Wnt, nuclear factor-kappa B (NF-κB), and Notch [1,8,9].

**Figure 1.** Overview of biological activity and applications of resveratrol.

RSV is being studied as a cancer prevention agent; it is effective both in vivo and in vitro because it interferes with the cellular activities linked to the stages of cancer development: initiation, promotion, and progression. It exhibits antioxidant, anti-mutagenic, and anti-inflammatory properties. Furthermore, RSV inhibits cyclooxygenase (COX)-1 and COX-2 enzymes, which are involved in converting arachidonic acid (AA) into substances that promote inflammation, potentially stimulating tumor growth and weakening immune defenses. The two COXs can activate carcinogens to reactive forms that produce cytogenetic damage. In addition to COX and hydroperoxidase inhibition, RSV induces the differentiation of human leukemic promyelocytes [1,7,8].

RSV induces the expression of phase II enzymes of the metabolism of active principles, the activity of quinone reductase (involved in detoxification), and inhibits ribonucleotide reductase, an enzyme complex that catalyzes the reduction in ribonucleotides to deoxyribonucleotides necessary for deoxyribonucleic acid (DNA) synthesis, cytochrome P450 CYP1A1, DNA polymerase, COX, and nicotinamide adenine dinucleotide reduced form (NADH)–ubiquinone oxidoreductase. The influence of RSV on these enzyme systems explains its antioxidant, antimutagenic, and anticarcinogenic potential [1,8].

RSV has a structure similar to that of diethylstilbestrol and, therefore, could exhibit a phytoestrogenic effect. Phytoestrogens have numerous physiological effects in humans, and the moderate consumption of RW at meals is useful for protecting the body against diseases and improving the aging process [10]. In vegetarians, but also in the case of Asians, people with a diet rich in phytoestrogens, there is a low incidence of breast, endometrial, ovarian, and prostate cancer (PC). Phytoestrogens are natural, non-steroidal plant components structurally and functionally similar to estrogens produced by the human body. Studies have shown that these compounds alleviate menopause symptoms and have anticancer properties, conferring protection against osteoporosis and hormone-dependent cancers (breast, prostate). In addition, phytoestrogens inhibit angiogenesis and cell cycle progression [11]. Phytoestrogens bind and activate estrogen receptors (ERs) but are less effective than estrogens of endogenous origin. RSV exerts its antiestrogenic activity, competing with 17β-estradiol for binding to ERs. By antagonizing the stimulatory effect of estradiol, RSV inhibits the growth of ER-positive human breast cells, acting both at the cellular level (cell growth) and at the molecular level (gene activation) [12,13].

Research has highlighted the fact that moderate intake of wine can have beneficial effects on human health, attributed especially to phenolic derivatives. In addition to flavonoids, polyphenolic acids (e.g., hydroxybenzoic acids, hydroxycinnamic acids) and stilbenes are important compounds present in grapes and wine. The remarkable properties of wine are due to polyphenolic compounds, mainly through their antioxidant activity and ability to eliminate free radicals. In addition, these compounds [(+)-catechol, (−)-epicatechol, proanthocyanidins, quercetin, kaempferol, RSV, gallic acid, caffeic acid] have been reported to exhibit a multitude of pharmacological actions, including cardioprotective, antiatherogenic, anti-inflammatory, antitumor, antibacterial, antiviral. From a phytochemical point of view, grapes represent a natural combination of polyphenols with a synergistic effect in the prevention of diseases and maintaining the health of the body. Epidemiological studies and clinical trials have shown that a regular and moderate consumption of RW (1–2 glasses/day) is associated with a low incidence of CVDs, diabetes, and some types of cancer [14–16].

The bioavailability of polyphenolic compounds varies within quite wide limits, so the most abundant polyphenols in wine are not necessarily the ones that determine the highest levels of active metabolites in the target tissues. Since wine is a complex mixture, it is probable that various chemical components, primarily polyphenols and their metabolites, work together to positively impact human health [14–16]. RSV is more lipophilic than other phenolic compounds, potentially enhancing its cell permeability and absorption. However, its bioavailability is low due to fast metabolism, resulting in discrepancies between in vitro and in vivo results [2,3].

This paper reviews the evidence supporting the beneficial effect of RSV for various pathological conditions, such as neoplastic diseases; neurodegeneration; brain injury; metabolic syndrome (MetS); diabetes; obesity; non-alcoholic fatty liver disease (NAFLD); CVDs; immune disorders; and bacterial, viral, and fungal infections. In addition, the study focused on the ultra-high-performance liquid chromatography–mass spectrometry (UHPLC–MS) analysis of tRSV in Romanian wine samples, both white and red varieties, providing a comprehensive overview of tRSV content across different types of wine.

## 2. Natural Occurrence of Resveratrol and Related Compounds

RSV has been identified in numerous natural resources of plant origin, e.g., grapevines, *Vitis vinifera* L. and *V. labrusca* L. (*Vitaceae*), the most important source; the roots of *Polygonum cuspidatum* Sieb., Japanese knotweed (*Polygonaceae*); the rhizomes and roots of *Holoschoenus vulgaris* Link. sin. *Scirpus holoschoenus* L., marsh cypress (*Cyperaceae*); the roots of *Veratrum grandiflorum* (Maxim. ex Baker) Loes., white sedge, and *V. formosanum* O. Loes. (*Liliaceae*), an endemic species in Taiwan; the leaves of *Eucalyptus globulus* Labill. and *E. accedens* W. Fitzg., eucalyptus (*Myrtaceae*); peanuts, *Arachis hypogaea* L. (*Fabaceae*) and peanut butter; the leaves of spruce, *Picea abies* L. (*Pinaceae*); the fruits of *Vaccinium myrtillus* L., such as blueberry, *V. corymbosum* L., American blueberry, and *V. macrocarpon* Ait., and American cranberry (*Ericaceae*); cocoa beans, *Theobroma cacao* L. (*Sterculiaceae*) and chocolate; blackberry, *Rubus fruticosus* L. (*Rosaceae*); tomatoes, *Lycopersicon esculentum* Mill. (*Solanaceae*); the aerial part of *Pterolobium hexapetallum* (Roth) Santapau and Wagh (*Fabaceae*); the fruits and leaves of *Cassia garrettiana* Caraib. and *C. quinquangulata* Rich. (*Fabaceae*); the wood of *Bauhinia racemosa* Lam. (*Fabaceae*); sweet almonds, *Prunus amygdalus* Bartock sin. *Prunus dulcis* (Mill.) D. A. Webb (*Rosaceae*); different varieties of hops used mainly in the beer industry (Amarillo, Cascade, Nugget, Simcoe, Sterling, Tomahawk, Vanguard, Warrior, Wilhamette), *Humulus lupulus* L. sin. *Lupulus amarus* Gilib. (*Cannabaceae*); apples, raspberries, plums, pistachios, hazelnuts; and also, in the products derived from the processing of these raw materials (such as natural juices/syrups, various hydroalcoholic extracts, wines) [2,3,5,7,9,17–19].

Spermatophytes, such as grapevines, generate RSV, which is the most important stilbenoid synthesized by plants, as a defense mechanism in response to microbial attack, toxins, or UV radiation, as a naturally occurring phytoalexin. *V. vinifera* L., mainly in the form of wine, is a valuable dietary source of stilbenoids, such as tRSV, which has the most important anti-aging biological effects in in vitro and in vivo studies. Hence, the compounds found in highly resistant strains are essential for the development of resistant crops, natural spray reagents, and novel nutritional supplements or phytopharmaceuticals [3,7].

RSV is found in large quantities, especially in the skin, but not in the pulp of the grapes. RSV content in wines obtained from different grape varieties is extremely variable. Typically, white wines (those produced from the White Burgunder, Riesling, Ortega, and Gewürztraminer varieties) contain about 10 times less RSV than RWs (from the Cabernet Mitos, Cabernet Cubin, Syrah, Spätburgunder/Pinot noir, Cabernet Sauvignon and Merlot). In RW, tRSV amounts generally vary between 0.1 and 15 mg/L. RSV concentration is influenced by several factors, such as grape variety, genotype, climate zone, soil type, exposure to light, pest control, agrotechnical works, maturity, post-harvest treatment, storage, and preservation [2,17].

The presence of RSV as a bioactive compound was first reported in eight Western Romanian propolis samples, along with kaempferol, quercetin, and rosmarinic acid. The principal component analysis showed clustering of the propolis samples, according to the polyphenolic profile similarity [20].

The structure of phenanthrene derivatives, tri-O-methyl-RSV, methyl-tri-O-methyl-gallate, pterostilbene, RSV, and methyl-gallate were determined by analytical and spectroscopic methods from the stems of *P. hexapetallum* [19]. Bioassay-directed purification of the active compounds from an ethanol extract of *Psoralea corylifolia* L., babchi (*Fabaceae*)

led to the isolation of corylifolin and bakuchiol as DNA polymerase inhibitors and also, based on their structures, RSV was tested and demonstrated to be effective as a DNA polymerase inhibitor [21].

Viniferin is a derivative of RSV, which has several isomers, e.g., $\alpha$-viniferin, $\beta$-viniferin, $\gamma$-viniferin, $\delta$-viniferin, $\varepsilon$-viniferin, R-viniferin (vitisin A), R2-viniferin (vitisin B). $\delta$-Viniferin is a dehydrodimer of RSV, an isomer of $\varepsilon$-viniferin. It occurs in vitro by oxidative dimerization of RSV by plant peroxidases or fungal laccases. Also, $\delta$-viniferin was recently identified in wines, in cell cultures, and in grapevine leaves infected with *Plasmopara viticola* Berk. and M.A. Curtis (*Peronosporaceae*) or irradiated with UVC. $\delta$-Viniferin, identified in higher concentration than $\varepsilon$-viniferin, is one of the most important phytoalexins derived from RSV. Viniferins have a series of important biological activities, such as anti-inflammatory, anti-psoriatic, anti-diabetic, anti-parasitic, anti-cancer, anti-angiogenic, antioxidant, anti-melanogenic, neuroprotective, antiviral, antibacterial, antifungal, anti-diarrheal, anti-obesity [22,23].

Pallidol is a natural dimer of RSV, also identified in RW, and has antioxidant and antifungal properties. At low concentrations, pallidol has strong singlet oxygen inactivation effects and is ineffective in eliminating hydroxyl radicals or superoxide anions. Kinetic studies showed that the reaction of pallidol with singlet oxygen had a very high constant. Therefore, pallidol could be used as a pharmacological agent in singlet oxygen-mediated diseases, thus contributing to the beneficial health effects of RW [24].

Hopeaphenol, identified for the first time in RW, especially from the Merlot variety, has a stilbenoid tetramer structure. It was determined by reversed-phase high-performance liquid chromatography (RP–HPLC) analysis of wine samples from North Africa (Algeria), alongside the known stilbenoid components: *t*RSV, *trans*-piceid, *trans*-$\varepsilon$-viniferin, pallidol [25].

## 3. Antitumor Properties

Chemoprevention by non-toxic agents, such as RSV, has been demonstrated to provide protection against various cancer types. Nevertheless, the molecular mechanisms responsible for anti-tumorigenic or chemopreventive actions of this phytochemical continue to be mostly unidentified [26,27].

### 3.1. Leukemia and Lymphoma

RSV has been determined to inhibit carcinogenesis in murine models. Besides its chemopreventive action, it also induces apoptotic cell death in human leukemia cells, exhibiting specific involvement of the cluster of differentiation (CD)95-CD95L system in the antitumor activity of RSV and emphasizing the chemotherapeutic ability of this natural compound [28]. It is a significant inhibitor of ribonucleotide reductase and DNA synthesis in mammalian cells, with possible utilizations as an antiproliferative and chemopreventive factor [27,29].

Studies demonstrate that RSV can inhibit every step of multistage carcinogenesis by reducing the viability and DNA synthesis ability of promyelocytic leukemia cells, inducing apoptotic cell death, and decreasing the expression of anti-apoptotic B-cell lymphoma-2 (Bcl-2) [30]. In the concentration spectrum of 20 $\mu$M and higher, it blocked the S-phase and caused apoptosis in the T-cell acute lymphoblastic leukemia (T-ALL), insufficient in functional p53 and p16 [31]. Other studies had the same results in human histiocytic lymphoma cells, indicating that RSV is an important cell cycle blocker in addition to being a chemopreventive agent [27,32].

In the case of acute monocytic leukemia, researchers have shown that RSV inhibits the growth of Tohoku Hospital Pediatrics-1 (THP-1) cells by inducing apoptosis, independent of the Fas/FasL signaling pathway and has no toxic effect on differentiated THP-1 cells [33]. Other data established that RSV and its correspondent, piceatannol, are powerful generators of apoptosis in BJAB Burkitt-like lymphoma cells, with a median effective dose of 25 $\mu$M, autonomous of the CD95/Fas signaling cascade [34].

*t*RSV, ε-viniferin, and vineatrol induced apoptosis in chronic leukemic B cells by activating caspase-3, reducing the mitochondrial transmembrane potential, and inhibiting both the expressions of the Bcl-2 anti-apoptotic protein and the inducible nitric oxide synthase (iNOS) [35,36]. Some studies suggest the involvement of apoptosis signal-regulating kinase 1/c-Jun N-terminal kinase (JNK) signaling in the regulation of FasL expression and following apoptosis induced by RSV in leukemia cells [37].

The inhibitory effect on hydrogen peroxide ($H_2O_2$)-induced apoptosis is attributed to the antioxidant property of RSV, which, in small concentrations, can inhibit death signaling in human leukemia cells via nicotinamide adenine dinucleotide phosphate reduced-form (NADPH) oxidase-dependent increase in intracellular superoxide that stops mitochondrial $H_2O_2$ synthesis [36,38].

Researchers determined that RSV has the ability to sensitize cancer cells to X-irradiation. The study involved cervix carcinoma, chronic myeloid leukemia, and multiple myeloma and demonstrated that the addition of RSV alone induced apoptosis and inhibited cell growth, while concomitant treatment with either RSV or X-irradiation caused a synergical reaction [39].

A plasma membrane receptor for RSV has been discovered near the arginine–glycine–aspartate recognition site on integrin $\alpha v \beta 3$ that participates in the proapoptotic mechanism of RSV in tumor cells, along with stimulating the intranuclear COX-2 accumulation, correlated to the activation of p53 [40].

The possible antitumor actions of RSV against the natural killer (NK) cell malignancies, especially aggressive NK cell leukemias and lymphomas, include strong G0/G1 cell cycle blockage and substantial cell proliferation and apoptosis restriction. Furthermore, RSV caused synergistic activity on the apoptotic and antiproliferative effects of L-asparaginase against cancer cells [41].

RSV's effectiveness varies among different human cancer cells, showing greater potential to induce cell death in leukemic cells than in solid tumor cells by inhibiting Akt activation through Ras downregulation [42].

### 3.2. Glioma, Glioblastoma and Neuroblastoma

Neuroblastoma is an aggressive type of tumor that requires a more efficient and less cytotoxic treatment. Studies indicate that the activity of RSV in neuroblastoma cells can be mediated by functionally activated p53 [43]. RSV intervenes in the proliferation, apoptosis, and cell cycle modification of neuroblastoma cells, inducing important cytotoxicity and increased apoptosis. S-phase accumulation is connected to the downregulation of p21 and up-regulation of cyclin E [44].

During an investigation in rat gliomas, RSV exerted important glioma cell cytotoxicity and apoptosis, induced antitumor effects on the spinal cord and intracerebral gliomas, and inhibited angiogenesis in spinal cord gliomas [45]. RSV elevates cellular cytotoxicity and inhibits the proliferation of rat neuroblastoma cells by activating mitochondria-mediated intrinsic caspase-dependent pathway [36,46].

Human glioma cells were utilized to comprehend the molecular mechanisms of the anticancer effect of RSV. The phytocompound caused apoptosis of glioma cells, as evaluated by lactate dehydrogenase release and internucleosomal DNA fragmentation analysis. RSV activated caspase-3, which leads to cytochrome c release from mitochondria to the cytoplasm and activation of caspase-9 [36,47].

RSV has the ability to suppress the growth and proliferation of glioma and improve its apoptosis through upregulating the leucine repeat immunoglobulin-like protein 1 (LRIG1) gene expression, which stimulates antiglioma growth, suggesting that LRIG1 is a new biological objective in antiglioma cell proliferation and growth [48].

Researchers investigated autophagy induced by RSV in human glioma cells, which inhibited growth and induced cell apoptosis. Glioma cells firmly expressing green fluorescent protein (GFP) merged to LC3, engaged more GFP–LC3-labeled autophagosomes, and elevated the percentage of cells with GFP–LC3-labeled autophagosome. Also, RSV-

induced autophagy was regulated by P38 and the extracellular signal-regulated kinase 1/2 (ERK1/2) pathway [49].

Glioblastoma multiforme (GBM) is the most destructive intracranial tumor that over-expresses the YKL-40 gene, related to the tumor stages of human primary astrocytoma and glioma cell proliferation. It was confirmed that RSV inhibits YKL-40 expression by limiting the exertion of its supporter and decreasing messenger ribonucleic acid (mRNA) transcription and protein activity in vitro, involving the ERK1/2 pathway [50].

Invasion and metastasis of glioblastoma-initiating cells are accountable for the evolution and recurrence of GBM. In vitro and in vivo observations suggest that RSV inhibits the adhesion, invasion, and migration of glioblastoma-initiating cells. Furthermore, it suppresses the invasion of glioblastoma-initiating cells by the inhibition of PI3K/Akt/NF-κB signal transduction and the following suppression of MMP-2 expression [51].

Recent findings exhibited the effect of RSV for the amelioration of glioblastoma inflammatory response by reducing NOD-like receptor protein 3 (NLRP3) inflammasome activation through inhibition of the Janus tyrosine kinase 2/signal transducer and activator of the transcription 3 (JAK2/STAT3) pathway [52].

### 3.3. Meningioma

RSV may be effective against meningioma cells by curbing their growth and triggering apoptosis. It increases cleaved caspase-3 protein levels while reducing pro-caspase-3 protein and Bcl-2 mRNA levels. Additionally, the RSV mechanism includes the upregulation of miR-34a-3p, which targets the 3′-untranslated region (3′-UTR) of Bcl-2, decreasing its protein levels and thereby promoting apoptosis in these cells. Therefore, the RSV anti-proliferative and pro-apoptotic effects in meningioma cells are mediated through the regulation of miR-34a-3p and subsequent suppression of Bcl-2 [53].

### 3.4. Estrogen-Dependent Pituitary Tumor

Phytoestrogens have been widely studied due to their capacity to reduce the incidence of various estrogen-dependent tumors. Some of them were proven to be active in pituitary tumor cells by binding to the ERs, stimulating growth, and causing an estrogenic response, such as prolactin secretion. Nevertheless, RSV does not attach to the ERs. Zearalenone, genistein, and coumestrol have been demonstrated to induce prolactin secretion and to encourage growth, while RSV exhibited prolactin secretion but without growth stimulation [54].

### 3.5. Breast Cancer

Breast cancer (BC) is one of the most prevalent tumors in females and is culpable for the principal cancer-related deaths after lung cancer. The anticancer mechanisms of RSV concerning BC involve the restriction of cell proliferation, decreasing of cell viability, invasion, and metastasis. However, despite its favorable therapeutic effects, it has limitations due to the reduced pharmacokinetics (PK), stability, and poor water solubility, implying the help of nanodelivery systems to enhance its bioavailability [55,56]. RSV-modified mesoporous silica nanoparticles (NPs) inhibit BC progression more efficiently, in comparison to the phytochemical alone, via suppressing the NF-κB signaling pathway, indicating that they are a novel and more secure method for applying in the complex therapeutics of BC [57].

Some research involved RSV-loaded solid lipid NPs that were properly designed to improve BC cells. The NPs presented a higher capacity than free RSV in suppressing the proliferation of tumor cells and displayed much better inhibitory actions on the incursion and migration of BC cells [58,59]. Additional data, important in the BC applications, established that transferrin–cathepsin B cleavable peptide-modified mesoporous silica NPs encapsulated RSV substantially reduced cell viability and enhanced cell apoptosis [60].

Research has shown that RSV activates the transcription of cytochrome P450 CYP1A1 and reduces the formation of carcinogen-induced preneoplastic lesions in mouse mammary tissues. It also prevents the growth of mouse skin tumors in two-stage models, inhibits the enzymatic actions of COX-1 and COX-2 in cell-free experiments, and blocks COX-2 mRNA along with the activation of protein kinase C (PKC) and activator protein-1 (AP-1)-mediated gene expressions in mammary epithelial cells [61].

A possible mechanism by which RSV could control the cell cycle and apoptosis has been investigated, exploring the human BC MCF-7 cell line. The results showed a dose-dependent anti-proliferative effect of RSV that was correlated with a notable inhibition of cyclin D and cyclin-dependent kinase (CDK)4 proteins and stimulation of p53 and CDK inhibitor p21$^{WAF1/CIP}$ [62,63]. Apoptosis caused by RSV was determined to happen exclusively in BC cells exhibiting wild-type p53 but not in mutant p53-expressing cells [64].

The growth factor heregulin-beta1 (HRG-β1) promotes the tumorigenicity and metastasis of human BC cells. MMP-9 is a crucial enzyme in the degeneration of extracellular matrices and has an altered expression in BC infiltration and metastasis. RSV showed anticarcinogenic activities in both in vitro and in vivo explorations that imply it suppresses the MMP-9 expression in BC cells, connected with the downregulation of the mitogen-activated protein kinase (MAPK)/ERK signaling pathway [65].

Metastasis is an important cause of decease in patients with BC. In the evolution of the tumor, epithelial–mesenchymal transition (EMT) is essential to supporting the invasion and migration of cancer cells. RSV suppresses the migration of human BC cells by invalidating the transforming growth factor-beta1 (TGF-β1)-induced EMT and inhibits lung metastasis in a xenograft-bearing mouse model [66]. The cytotoxic action of RSV on human BC cells may be correlated with its pro-oxidant effect, which inhibits casein kinase 2 activity, influencing cell viability and mitochondrial function [36,67].

*3.6. Hepatocellular Carcinoma*

Many studies are designed to examine the chemopreventive action of various doses of RSV against hepatocellular carcinoma (HCC). It was demonstrated that a 100 mg/kg dose of RSV can be a suitable treatment for HCC produced by diethylnitrosamine in rats [68]. Other evidence revealed that RSV suppresses tumor necrosis factor-alpha (TNF-α)-regulated MMP-9 expression and progression of human HCC cells, which correlates with the inhibition of the NF-κB signaling axis [69]. RSV reduces cell proliferation, produces reactive oxygen species (ROS), and generates DNA single-strand breaks. Due to its ability to regulate the expression of proteins implicated in the redox pathways and apoptosis, RSV induces hepatic cancer cell death by inhibiting the expression of antioxidant proteins [70].

It was demonstrated that several natural polyphenols stimulate the activity of telomerase. Recent studies show that silent information regulator 1 (SIRT1) and nuclear factor erythroid 2 (NF-E2)-related factor 2 (Nrf2) are implicated in the modulation of human telomerase reverse transcriptase (hTERT). RSV, together with kuromanin chloride and gallic acid, stimulates the hTERT gene expression in the hepatocellular tumor cells, probably by activation of the SIRT1/Nrf2 signaling cascade. Accordingly, by addressing hTERT or this axis, the contra-tumor action of this class of phytochemicals might be increased [71].

When treated with a mixture of bioactive substances, RSV and berberine, human HCC cells, Hep-G2 and Hep-3B, with various p53 status, showed distinct reactions in cell viability by p53-reliant apoptosis pathway activation. An efficient concentration of berberine, combined with RSV, could be decreased even to 50% in tumor therapy and substantially decrease the viability of wild-type p53 Hep-G2 and null p53-mutant Hep-3B. Therefore, p53 status in hepatocellular tumor cell lines regulates the reactions to phytotherapy [72].

### 3.7. Pancreatic Carcinoma

Pancreatic tumor, one of the most frequent causes of cancer-connected mortality, has a weak outcome, brief survival rate, and resistance to therapy. Studies have shown that RSV has chemopreventive and treatment action in pancreatic cancer: it reduces the propagation of pancreatic cancer cells; generates apoptosis and cell cycle arrest; prevents metastasis and infiltration of pancreatic cancer cells; obstructs the reproduction and viability of tumor stem cells; and increases the chemoradiosensitization of cancer cells [27,73].

The examination of the mechanisms implicated in the in vitro antitumor activity of RSV revealed that it suppresses the proliferation of pancreatic cancer cells, which, after incubation with RSV, leads to cell apoptosis and cell cycle cessation. RSV generates the activation of caspases and modulates the activity of the contra-apoptotic proteins Bcl-xL, Bcl-2, and X-linked inhibitor of apoptosis protein (XIAP) and the proapoptotic protein Bax [74].

RSV plays two roles in pancreatic tumors: it inhibits cancer by increasing Bax expression but also promotes cancer by boosting vascular endothelial growth factor-B (VEGF-B). However, the cancer-inhibiting effects of RSV are stronger than its cancer-promoting ones. Therefore, combining RSV with a VEGF-B inhibitor could offer an effective treatment for pancreatic cancer [75]. Additionally, the polydopamine-based nanomedicine containing RSV and hyaluronidase suppressed the invasive behavior of pancreatic cancer cells in a tumor sphere model enriched with hyaluronic acid [76].

Hyperlipidemia acute pancreatitis is an important undercover threat concerning our health. Hence, researchers analyzed the preventive influence and the intrinsic mechanism of RSV in hyperlipidemia acute pancreatitis in mice and discovered that the RSV pretreatment efficiently suppresses pancreatic pathological injury and inhibits inflammation and oxidative stress by restraining the NF-κB signaling pathway and improving intestinal microbiota [77].

### 3.8. Esophageal Carcinoma

The purpose of several studies was to investigate the probability and possible mechanism of the inhibition of N-nitrosomethylbenzylamine-induced rat esophageal tumorigenesis by RSV. It substantially reduces the up-regulated expression of COX-1 and COX-2 and the enhanced levels of prostaglandin (PG)E2 synthesis, and for that reason, it can be a promising natural anti-carcinogenesis factor for the prevention and therapy of human esophageal cancer [78].

In vitro experiments were conducted to evaluate the RSV-induced apoptosis in esophageal cancer cells and its connection to the expressions of Bcl-2 and Bax. They concluded that RSV is capable of generating apoptosis in esophageal tumors by influencing the expressions of these two apoptosis-modulated genes: down-regulating Bcl-2 and up-regulating Bax [79].

### 3.9. Gastric Cancer

Gastric cancer (GC) is one of the most frequent tumors with serious malignancy. Numerous studies have determined the implication of Wnt, Nrf2, MAPK, and PI3K/Akt signaling pathways in this type of cancer. RSV involves several signaling pathways to generate apoptotic cell death, along with obstructing the migration and metastasis of GC cells [80].

Some observations showed that RSV is capable of decreasing viability and generating apoptosis in GC cells by suppressing NF-κB activation [81]. Other studies established that RSV elevates superoxide dismutase (SOD) activity but reduces NF-κB transcriptional activity and heparanase enzymatic activity, which corresponds with the diminishing of invasion potential in GC cells [82].

It has also been reported that RSV inhibits the proliferation, migration, and invasion of human gastric tumor cells through the metastasis-associated lung adenocarcinoma

transcript 1 (MALAT1)/micro-ribonucleic acid (miRNA)-383-5p/DNA damage-inducible transcript 4 pathway and generates apoptosis [83]. Overexpression of miRNA-155-5p, which generates oncogenesis in gastric tumors, is regulated by RSV, one of the mechanisms that confirms the important antiproliferative activity of this phytochemical [84].

Chemotherapy toxicity is not principally caused by the medication itself but is generated via cell-free chromatin particles that are discharged from chemotherapy-induced decaying cells and are immediately internalized by healthy cells. At this point, they produce double-stranded DNA (dsDNA) breaks and stimulate inflammatory cytokines. It has been discovered that they can be inactivated by oxygen radicals that are induced after treatment with RSV and copper and consequently curtail chemotherapy toxicity [85].

### 3.10. Colorectal Carcinoma

Chemotherapy is universally used for treating colorectal tumors. Because of the high toxicity to normal cells and the emerging cancer resistance, it is preferable to use chemotherapy combined with natural bioactive agents that can assuage the side effects as coadjuvants. It was confirmed that the co-administration of RSV along with chemotherapy has a higher anti-proliferative effect; elevates chemotherapy-induced cytotoxicity; and synergistically has the ability to sensitize tumor cells to chemotherapy as a result of its oxidative, proapoptotic, and anti-inflammatory action [86].

An RSV tetramer, vaticanol C, was identified in the stem bark of *Vatica rassak* (Korth.) Blume, resak (*Dipterocarpaceae*), substantially inhibits cell proliferation via initiation of apoptosis, which was represented by nuclear alterations and DNA ladder development in several distinct human colon cancer cell lines [87]. Another phytocompound, *t*RSV, has an antiproliferative effect without cytotoxicity and generates apoptosis in human colorectal carcinoma through a ROS-dependent apoptosis pathway [88].

Certain studies submitted that the antitumor activity of RSV is viable by activating Hippo/YAP signaling and moderately perturbing the YAP–TEAD interaction, is very useful information in colon cancer therapy [89]. Reports documented that 0.5 g or 1.0 g of RSV per os every day generates efficient concentrations in the human digestive tube to induce antiproliferative activity, showing great potential as a colorectal cancer chemopreventive agent [90]. The anti-proliferative effect of a standardized combination of compounds: *t*RSV, quercetin, vitamin E, and selenium (Neoplasmoxan) was reported in mouse colorectal carcinoma, suggesting that this preparation may be a synergic treatment in addition to standard chemotherapy or radiotherapy [91].

When the comparative analysis of the anti-proliferative, proapoptotic, and oxidative stress potential of RSV–zein NPs versus free RSV was performed against human colorectal carcinoma cells, the following observations were made: the NPs achieved cell cycle suppression promoted by enhanced cytotoxicity, response, and oxidative stress markers concentrations than RSV alone. These data suggest that the chemopreventive profile of RSV is expanded as a result of the efficacious delivery system using histocompatible nano-dispersion [92].

### 3.11. Lung Cancer

The antitumor effect of RSV has been connected to transformations in the sphingolipid metabolism. The analysis performed on the enzymes accountable for the aggregation of the two most relevant sphingolipids, apoptosis-promoting ceramide and proliferation-stimulating sphingosine-1-phosphate (S1P), in human lung adenocarcinoma cells, revealed that RSV administration enhanced apoptosis-promoting ceramide and sphingosine and reduced sphingomyelin and S1P [93].

Studies show that RSV generated apoptosis in lung cancer cells by precisely targeting phosphoAkt (pAkt) and cellular FLICE-like inhibitory protein (c-FLIP) downregulation as a result of proteasomal degradation contingent to epidermal growth factor receptor (EGFR) [94]. Other data determined that it inhibits lung tumor cell proliferation by suppressing COX-2 expression [95]. New mechanisms of RSV tumor suppression have been

discovered, implying that this phytochemical has the ability to reprogram M2-like tumor-associated macrophages and tumor-infiltrating CD8T effector cells [96].

Further investigations reported the implication of autophagy in RSV-induced apoptosis and its possible molecular mechanisms via $Ca^{2+}$/adenosine monophosphate-activated protein kinase (AMPK)–mammalian target of rapamycin (mTOR) signaling pathway in human lung adenocarcinoma cells [97]. In a non-small-cell lung cancer assay, RSV-induced autophagy and apoptosis at levels higher than 55 μM were mediated by the nerve growth factor receptor–AMPK–mTOR signaling pathway [98].

There are numerous indications that cancer stem cells are a crucial conductor of carcinogenesis and tumor reoccurrence, involving cytokines and chemokines—the cellular and soluble constituents of the tumor microenvironment. The expression level of interleukin (IL)-6 is interconnected to the existence of lung cancer stem-like cells in human lung carcinoma. RSV has the ability to multi-target lung cancer stem-like cells and IL-6 in the tumor microenvironment, and it is an innovative approach to the therapy and prevention of lung cancer [99].

Regarding the lung cancer metastasis of melanoma, the cytotoxicity of RSV was assessed in immunocompetent mice. The in vitro development of melanoma cells was substantially inhibited by RSV. The average survival rate of mice was improved, and the lung tumor proliferation was suppressed through in vivo intraperitoneal injection of 40 mg/kg RSV. The lung tumors exhibited enhanced interferon-gamma (IFN-γ) and C-X-C motif chemokine ligand 10 (CXCL10) levels and reduced angiogenesis and infiltration [100].

### 3.12. Bladder Cancer

Relevant information about the influence of RSV on human bladder carcinoma cells discloses that its effects are dose-dependent and may encourage the creation of impending chemoprevention approaches. Elevated doses of RSV produce apoptosis of bladder tumor cells, while reduced dosage provides protection by regulating Bcl-2 protein during oxidative stress conditions [27,101].

When the effects of RSV were evaluated in two human renal cell carcinoma cell lines, the findings indicated that it inhibits cell proliferation, migration, and infiltration due to deactivation of the Akt and ERK1/2 signaling cascades, depending on concentration and time [102].

Recent studies demonstrated the antiproliferative action of RSV in bladder tumor cells, mediated by ROS production, intervention in the cell cycle, and suppression of cell migration, regardless of their tumor protein p53 (TP53) status. The enhanced ROS formation after RSV administration was concluded by decreased cell viability and proliferation in all bladder cancer cell lines [103].

### 3.13. Cervical Carcinoma

Cervical cancer is one of the most prevalent tumors in women, especially in developing countries. In order to find more affordable approaches in therapy, many studies included plant polyphenols, such as RSV or pterostilbene, which have presented antitumor and anti-human papillomavirus (HPV) effects as opposed to cervical cancer cells. They reduce clonogenic survival, diminish cell migration, block cells at the S-phase, and decrease the number of mitotic cells. The data confirmed a mechanism that concerns the downregulation of the HPV E6 oncoprotein, re-establishment of functional p53 protein, and stimulation of apoptotic pathways, with pterostilbene being more effective compared to RSV [104].

RSV suppresses proliferation and generates apoptosis in human cervical cancer cells in a dose- and time-relative process. The RSV-inflicted cell shrinkage and apoptosis are followed by the stimulation of caspase-3 and caspase-9, upregulation of the proapoptotic Bcl-2-associated X protein, and downregulation of the contra-apoptotic genes Bcl-2 and Bcl-extra-large. Moreover, p53, an indispensable protein for cell survival and cell cycle

continuity, presents RSV-induced increased expression levels in cervical carcinoma cells [105].

Currently, it has been concluded that RSV causes apoptosis and inhibits the migration and invasion of cervical tumor cells by inhibiting the Hedgehog (Hh) signal pathway. This natural compound downregulates the expression levels of Hh signal pathway proteins (smoothened, zinc finger transcription factors, and sonic Hh homolog). The Hh agonist purmorphamine inverts the effects of RSV [106].

### 3.14. Ovarian Cancer

Epithelial ovarian carcinoma constitutes the deadliest gynecological tumor and a considerable health problem because of the serious mortality percentage. The potential antitumor action of RSV was appraised concurrently in in vitro and in vivo ovarian cancer. Even though it was efficient in inhibiting the in vitro cellular invasion of ovarian cancer cells, it was inactive in in vivo ovarian cancers in rats [107].

However, when the anti-cancer mechanism of RSV was evaluated in human ovarian cancer cells, it was validated that RSV generates ROS, which provokes autophagy and following apoptosis. RSV stimulates autophagy-related gene 5 (ATG5) expression and triggers LC3 cleavage. The RSV-induced apoptotic cell death was depreciated by pharmacological and also genetic suppression of autophagy by silencing RNA (siRNA)-targeting of ATG5. Therefore, it was confirmed that autophagy and RSV-induced apoptosis are correlated in human ovarian cancer [108].

### 3.15. Osteosarcoma

Osteosarcoma is a high-degree bone neoplasm with serious invasive capacity. It was reported that RSV suppresses proliferation and glycolysis, generates apoptosis, and decreases the invasiveness of osteosarcoma cells in vitro. In addition to this, RSV downregulates the expression of interconnected Wnt/β-catenin signaling pathway target genes and enhances the level of E-cadherin. In an extended-time administration of RSV, the expression of connexin 43 (Cx43) was enhanced. Therefore, these data confirm that the anticancer influence of RSV on osteosarcoma cells is achieved via up-regulating Cx43 and E-cadherin and inhibiting the Wnt/β-catenin signaling pathway [109,110].

Various studies investigated the role of RSV on proliferation and apoptosis in osteosarcoma cell lines and associated it with the activation of the SIRT1/silent information regulator 2 (SIR2) family of nicotinamide dinucleotide oxidized form ($NAD^+$)-dependent deacetylases, which influences the calorie restriction-regulated tumor inhibition. SIRT1 protein has an increased expression in cancer cells than in normal osteoblasts. RSV generates apoptosis in osteosarcoma cells conditional to the dosage but has an insignificant impact on normal osteoblasts. Moreover, the proapoptotic action of RSV may be stimulated by dietary limitation induced by L-asparaginase [111].

Recent examinations certified that RSV prevents cell proliferation, migration, and invasion, initiating apoptotic cell termination in osteosarcoma cells. The mechanism might be the downregulation of NF-κB and Akt intracellular signaling transduction. Additionally, the association with pyrrolidine dithiocarbamate, an NF-κB inhibitor, determined combined inhibition of osteosarcoma evolution [112].

### 3.16. Epidermoid Carcinoma and Melanoma

Nonmelanoma skin cancer is the most frequent malignancy caused by solar UV radiation, principally by its UVB segment (290–320 nm). RSV is a powerful antioxidant known as a "photochemopreventive agent" that can improve the damages provoked by UVB exposure to the epidermis. In a study on hairless mice, the topical application of RSV suppressed the UVB-induced effects, such as an increase in bifold skin thickness, skin edema, induction of COX and ornithine decarboxylase (ODC) enzyme activities, and protein expression of ODC, which are important factors in tumor development [113,114].

Regarding epidermoid carcinoma, the effect of RSV on transformed keratinocytes exhibiting mutant p53 was analyzed. It induced G1 cell cycle arrest by significantly suppressing G1 cell cycle regulatory proteins, including cyclins A and D1, CDK6, and p53-independent induction of p21$^{WAF1}$. RSV also down-regulated activating protein 1 DNA-binding and promoter activity, repressed MAPK/ERK1 or ERK1/2 signaling, and inhibited c-Jun [115].

With reference to melanoma treatment, the aim of various surveys was to investigate the aptitude of RSV to complement chemotherapy. In vitro RSV substantially reduces melanoma cell viability, targeting only the cancer cells and exempting fibroblast cell lines. In comparison to the administration of temozolomide alone, the treatment of malignant cells with 50 μM RSV and temozolomide for 72 h strongly augments cytotoxicity [116]. Western blot analysis confirmed that RSV generates the apoptosis of human melanoma cells by increasing the expression of Bcl-2-associated X protein and Bcl-2, presumably through the p53 pathway and stimulation of caspase-9 and caspase-3 [117].

In a study intended to evaluate the chemopreventive and antimutagenic action of RSV against skin neoplasm in vivo, croton oil-induced increase in ODC effects of dorsal epidermis cells in mouse and mouse ear edema model was used to highlight the anti-promotion activity of RSV. Also, 7,12-dimenthylbenz[*a*]anthracene (DMBA)/croton oil was utilized to cause mouse skin cancer. Administration of RSV reduced ODC activity in croton oil-induced dorsal epidermis. It also was demonstrated that RSV could inhibit mouse skin papilloma produced by DMBA/croton oil, which involves the expansion of cancer occurrence duration, reducing the prevalence of papilloma and decreasing tumor number per mouse [118].

### 3.17. Oral Squamous Cell Carcinoma

Oral squamous cell carcinoma is the most typical tumor of the head and neck. Even though surgery, radiation, and chemotherapy are adequate therapeutic protocols, the medication tolerance and toxicity of oral cancer patients continue to be an issue. Various approaches to understanding the intrinsic mechanisms of RSV-induced apoptosis were conducted on this type of neoplasm. In the case of human tongue squamous cell carcinoma, the inhibitory effect of RSV was possible along the mitochondrial pathway and the suppression of cell invasion and migration by down-regulating the EMT-inducing transcription factors [36,119].

In vitro studies suggest that RSV influences the adhesion, migration, and infiltration of oral squamous cell carcinoma cells; therefore, it has the ability to be a chemopreventive element for decreasing invasion and metastasis [120]. Other data corroborated that RSV alone or in association with quercetin, in concentrations similar to red wines, are efficient inhibitors of oral squamous carcinoma cell evolution and proliferation [121].

In combination with copper, RSV can reduce tumor genome instability and inflammation by inactivating the cell-free chromatin particles with the help of oxygen radicals. Regarding the advanced squamous cell carcinoma of the oral cavity, some results indicate that cell-free chromatin particles discharged into the tumor microenvironment from moribund malignant cells are universal generators for cancer manifestations and immune frontiers in remaining tumor cells. The capacity of this mixture to inactivate the chromatin particles presents the possibility of an innovative and non-toxic therapeutic plan, which, instead of exterminating the cancer cells, provides treatment via inhibiting immune checkpoints and tumor hallmarks [122].

### 3.18. Nasopharyngeal Cancer

The underlying mechanism of RSV has been investigated in various tumors. Regarding human nasopharyngeal carcinoma cells, it inhibits cell proliferation by suppressing Ki67 and proliferating cell nuclear antigen (PCNA) proteins and induces apoptosis by altering its analogous proteins, cleaved caspase-3 and cleaved caspase-9. RSV therapy diminishes the enhanced expression of survivin that inactivates its reaction on cell growth

and apoptosis. These results confirm that RSV can influence cell proliferation and demise in nasopharyngeal carcinoma cells [123].

Administration of RSV suppresses cell viability and generates apoptosis of the human nasopharyngeal carcinoma cells. Moreover, it was determined that RSV stimulates caspase-3 and modifies the Bax/Bcl-2 apoptotic signaling due to the up-regulation of AMPK activity and inhibition of p70S6K and S6 kinases, succeeding components of AMPK [124].

### 3.19. Prostate Cancer

PC is one of the most frequent kinds of tumors in men, related to the unrestrained growth of the prostate gland. The evolution of androgen-independent PC implicates fundamental ERK1/2 activation, promoted by the epidermal growth factor/transforming growth factor-alpha/EGF receptor pathway. It was ascertained that RSV inhibits ERK1/2 activation by altering PKC, specifically its isozyme PKC$\alpha$, in neoplastic human prostate epithelial cells in vivo [125].

The antitumor effect of RSV in PC can be correlated to the regulation of miRNAs, which are minuscule non-coding RNAs that antagonize gene expression. RSV down-regulates carcinogenic miRNAs and up-regulates cancer-inhibitor miRNAs in PC cells [126]. This mechanism may be elucidated by its capacity to inactivate (ERK1/2) Akt and reduce the concentrations of estrogen and insulin growth factor-1 receptor, suppressing the Akt/miR-21 signaling pathway [127].

New approaches support the benefit of RSV-loaded NPs for PC chemoprevention/chemotherapy due to the lack of adverse reactions on normal cells. Polymeric NPs encapsulating RSV were developed, and they generated substantially higher cytotoxicity to PC cells than free RSV, without negative cytotoxic repercussions on murine macrophages, even at 200 µM [128]. Similar results were obtained by the successful endeavor to expand systemic circulation and biological half-life of *t*RSV, utilizing solid lipid NPs to raise its antitumor ability [129].

The function of RSV as a radiosensitizer by targeting cancer stem cells in radioresistant PC cells was analyzed, employing stem cell markers and EMT markers, such as vimentin and E-cadherin. The association of RSV administration with ionizing radiation generated an important decline in cell viability and stem cell markers genes and altered EMT markers [130].

Vasculogenic mimicry is specifically correlated to tumor progression and metastasis, which is responsible for the poor prognosis of patients with PC. RSV decreases serum-induced invasion and vasculogenic mimicry formation in human PC cells. Blood-generated phosphorylation of erythropoietin-producing hepatocellular A2 and the expression of vascular endothelial (VE)-cadherin at the mRNA and protein positions are reduced by RSV. Blood-induced AKT signaling cascade, as well as metalloproteinase-2 and laminin subunit 5 gamma-2, are damaged by this phytochemical [131].

It is more and more acknowledged that the synergy of the glioma-associated oncogene axis and androgen receptor influences the evolution of PC. Recent in vitro and in vivo assays on xenograft models in mice confirmed that RSV suppresses the Hh signals and tumor necrosis factor receptor (TNFR)-associated factor 6 (TRAF6) levels to ameliorate PC progression. The mechanism of action involved EMT progression, reducing the tumor size and expressions of vimentin, VEGF, and MMP-7, as well as stimulating the expressions of E-cadherin and annexin 2 [132].

## 4. Neuroprotective Activity

### 4.1. Neurodegenerative Disorders

RSV, the polyphenol that can cross blood–brain barriers, has a powerful neuroprotective effect on various neurodegenerative diseases by mitochondria modulation mechanism. Mitochondrial function alteration is the most frequent cause and pathological

development in several neurodegenerative conditions, like amyotrophic lateral sclerosis, AD, Huntington's disease, or PD. The currently accessible therapies for neurodegenerative disorders are only symptomatic treatments and have undeniable toxic reactions. In conclusion, there is a strong necessity for a safe, natural, and efficient therapy protocol [133–135].

AD is an escalating and neurodegenerative pathology of the cortex and hippocampus, which ultimately leads to cognitive deterioration. Despite its ambiguous etiology, the existence of β-amyloid peptides in the learning and memory areas is an indication of AD. It has been demonstrated that RSV has antioxidant, anti-inflammatory, and neuroprotective effects and can reduce the toxicity and aggregation of β-amyloid peptides in the hippocampus, stimulate neurogenesis, and avoid hippocampal impairment. Moreover, the antioxidant effect of RSV is very effective in neuronal differentiation due to the activation of SIRT1. Because of the anti-inflammatory action, it inhibits M1 microglia activation, which plays a part in the induction of neurodegeneration and stimulates Th2 responses by enhancing anti-inflammatory cytokines and SIRT1 [136,137].

The mechanism and therapeutic action of RSV were explored in AD-like behavioral dysfunction caused by streptozotocin (STZ). The deteriorations in task-specific learning and memory provoked by the bilateral hippocampal microinjections of STZ were improved by RSV. The mechanism is correlated to the modulation of brain-derived neurotrophic factor expression and synaptic plasticity-associated proteins in the hippocampus [138].

### 4.2. Chemical-Induced Neurotoxicity

The aqueous extract of RSV from methanol extracts of the roots of *Vitis thunbergii* var. *sinuata* (Regel) Rehder (*Vitaceae*) showed substantial neuroprotection against glutamate-caused neurotoxicity in primary cultured rat cortical cells. By using fractionation processes, five RSV derivatives [vitisinols A, B, C, (+)-vitisin A and (+)-vitisin C] were isolated. Three of them presented the most important neuroprotective effect opposite to the neurotoxicity produced by glutamate, as recorded by cell viability of relatively 75–85%, at concentrations varying from 0.1 μM to 10 μM [139].

### 4.3. Scopolamine-Induced Cognitive Impairment

Researchers evaluated the role of RSV on cognitive disorders caused by scopolamine, a muscarinic antagonist, in mice. Scopolamine induced considerable extension of transfer latency on elevated plus maze, decreased step-down latency on a passive avoidance apparatus, and escape latency in Morris' water maze test, demonstrating cognitive deterioration. Unfortunately, RSV administration did not annul the effects of scopolamine and, therefore, did not improve the loss of cognitive functions in mice [140].

### 4.4. Resveratrol against Brain Injury

Traumatic brain injury is the most frequent type of craniocerebral injury that generates neurological deterioration and cognitive impairment. RSV recovers cognitive activity in neurological dysfunctions and senescent prototypes as a result of its anti-inflammatory effect. Synapse creation is crucial for cognitive function, and synaptophysin is one of the significant elements implicated in synapse induction. SIRT1 has a neuroprotective influence on several biological mechanisms, such as metabolism, inflammation, apoptosis, and autophagy. It was concluded that RSV enhances synaptophysin via activating the SIRT1/peroxisome proliferator-activated receptor-gamma coactivator-1 (PGC-1) cascade and promotes post-traumatic brain injury cognitive activity [141].

The influence of RSV against brain injury caused by radiation in rats was assessed. The radiation group of rats exhibited an important reduction in catalase (CAT), SOD, glutathione peroxidase, and glutathione reductase functions, including glutathione contents. The administration of RSV substantially improved these benchmarks in the brain tissues

of the animals. This result is connected to the capacity of RSV to target the free radicals, stimulate the activity of the antioxidant enzymes, enhance glutathione content, and inhibit the inflammatory reactions and apoptosis markers [142].

## 5. Metabolic Disturbances

### 5.1. Metabolic Syndrome

MetS is an important health issue correlated with enhanced risk factors for hepatic steatosis, which is the most frequent liver disorder nowadays. The protective actions of RSV over metabolic dysfunctions influenced by a high-fat, high-fructose diet were analyzed. The following observations were made on MetS rats: hepatic dysfunction, increased body weight, dyslipidemia, hepatic oxidative and inflammatory stress conditions, hepatic insulin resistance, stimulation of mRNA expression of sterol regulatory element binding protein 1c (SREBP-1c), and inhibition of mRNA expressions of peroxisome proliferator-activated receptor-alpha (PPAR-$\alpha$) and insulin receptor substrate-2 (IRS2). Treatment with RSV diminished liver steatosis, inflammatory state, and oxidative stress. Moreover, it ameliorated lipid profile, together with insulin reactivity and regressed modifications in hepatic mRNA expression levels of the investigated genes [143].

### 5.2. Antidiabetic Properties

Diabetic complications, such as cataracts and nephropathy, have numerous and complex mechanisms. RSV was investigated for its inhibitory effect versus rat lens and kidney aldose reductase in vitro, together with its capacity to block the generation of advanced glycation end products. The phytoalexin considerably amended glycemic status and renal function in diabetic rats, with an important reduction in the production of advanced glycation end products in the kidneys. This evidence highlights the capability of RSV as a potential therapeutic factor as opposed to long-term diabetic complications [144].

The protective effects of RSV treatment against diabetes and cardiovascular complications of diabetes are based on modulation of cell and molecular signaling pathways, such as improvement of insulin sensitivity, reduction in oxidative stress and inflammation, lipid metabolism regulation, increase in glucose transporter (GLUT) 4 expression, autophagy induction, regulation of SIRT1/AMPK axis [145].

In an experimental model of obese mice with type 2 diabetes mellitus (T2DM), 60-day RSV oral administration increased the nuclear amount of SIRT1 and restored glycemic control, as well as the normal values of GLUT2 (*Slc2a2* gene) and GLUT4 (*Slc2a4* gene) proteins in liver and muscle, respectively [146].

Quercetin and RSV have well-known favorable actions on diabetes and diabetic complications; nevertheless, the results of mixed treatment of these phytochemicals on diabetes are not completely elucidated. The combined treatment guarded the functions of hepatic glucose metabolic enzymes and the structure of pancreatic β-cells from diabetes, and it is remarkable that this cotreatment exhibited the most beneficial action on diabetic rats [147]. Other assays proved that the nutritional polyphenols RSV and piceatannol reduce postprandial hyperglycemia and suggest that suppression of intestinal $\alpha$-glucosidase function can be a possible mechanism associated with their antidiabetic effect [148].

Various clinical trials highlighted the beneficial effects of RSV treatment against diabetes. In a preliminary investigation on 13 patients with type 1 diabetes mellitus (T1DM), RSV 500 mg capsules supplementation, twice daily for 60 days, significantly decreased the values of fasting blood sugar (FBS) and hemoglobin A1c (HbA1c) compared to baseline. Moreover, in this exploratory clinical trial, RSV treatment also decreased the level of oxidative stress marker malondialdehyde (MDA) and increased the total antioxidant capacity (TAC) in T1D patients [149]. In a 4-week double-blind trial, which included 10 T2DM patients who received oral RSV treatment, 5 mg twice daily, a significant decrease in the homeostatic model assessment of insulin resistance (HOMA-IR) index and urinary excretion of *ortho*-tyrosine was found [150–152]. Also, compared with a placebo, in T2DM

elderly patients, RSV oral supplementation, 200 mg once daily for six months, significantly improved glucose homeostasis, HOMA-IR index, lipid profile, renal function, and inflammatory status by decreasing the level of proinflammatory cytokines [153]. In T2DM patients, RSV co-administration with oral hypoglycemic drugs for 24 weeks improved oxidative stress status, significantly decreasing plasma levels of MDA, high-sensitive C-reactive protein (hs-CRP), TNF-$\alpha$, and IL-6 [154,155].

### 5.3. Hyperlipidemia, Hepatic Steatosis, and Synaptic Impairment

In order to evaluate the impact of RSV on the lipid profile, a dose–response meta-analysis of randomized controlled trials was conducted. The evidence showed that the intake of RSV may substantially reduce the total cholesterol, triglyceride, and low-density lipoprotein (LDL)–cholesterol levels without modifying the level of high-density lipoprotein (HDL)–cholesterol. The decrease in LDL–cholesterol was more important in trials with an extended duration and in patients with T2DM, demonstrating that the dosage of the RSV treatment is a crucial element [156].

RSV produced via microorganism biotransformation exhibits better purity compared to the naturally obtained RSV, at a reduced expense than the chemically synthesized one. The beneficial actions of this novel product were examined in hamsters on a high-fat diet; liver enzymes, total cholesterol, triglycerides, serum glucose, and liver weight were substantially decreased. Moreover, HDL–cholesterol was also enhanced, while blood LDL–cholesterol was reduced. Triglycerides in liver tissue and fatty liver level were decreased, and memory-associated proteins, phosphorylation of calmodulin-related protein kinase II, and synaptophysin were enhanced in the cerebrum of the hamsters. The microorganism-biotransformed RSV has the potential to defend patients from hyperlipidemia, hyperglycemia, hepatic steatosis, and synaptic alteration [157].

### 5.4. Anti-Obesity and Anti-Atherosclerotic Effects

Atherogenic or high lipid nutrition is recognized to generate CVDs, and several studies explore the cardioprotective role of RSV. The treatment with RSV everted the anomalous blood lipid profile in a test involving atherogenic rats. Moreover, it improved the enzymatic and non-enzymatic antioxidants and stimulated the lipid metabolic proteins, demonstrating its beneficial role in lipid metabolism. Additionally, RSV repaired the tissue impairment and preserved the morphology of cardiac tissue. Therefore, RSV protects the cardiovascular system damaged by high lipid nutrition and is a favorable healthy lifestyle-adjuvant [158].

RWs are considered a great nutritional provenience for *t*RSV, but also for its derivatives, such as ε-viniferin, an RSV dimer that has a better anti-adipogenesis role in metabolic disorders. Furthermore, ε-viniferin is more efficient than *t*RSV due to its anti-obesity and anti-inflammatory actions in mice exposed to high-fat nutrition [159]. Other studies highlight that RSV, together with its dimers, ε-viniferin and δ-viniferin, are beneficial in restraining atherosclerosis via a related molecular mechanism, with distinct capability and competence [160].

Atherosclerosis is a pathological fundament of cardiovascular and cerebrovascular disorders, which involves signal transduction in its development. It was determined that RSV is very efficient in different stages of atherosclerosis evolution through modulating lipid metabolism, suppressing inflammatory reactions, and ameliorating oxidative stress via regulating the transforming growth factor/extracellular regulated protein kinases signaling axis [161].

### 5.5. Polycystic Ovary Syndrome

Polycystic ovary syndrome (PCOS) is the most frequent endocrinopathy concerning women of reproductive age, especially involving hyperandrogenism. Assays on individualized ovarian theca-interstitial cells indicated that RSV decreases androgen formation,

substantially inhibiting ovarian and adrenal androgens by stimulating insulin sensitivity and reducing the insulin level [162]. Recent investigations assert that RSV presents valuable phytoestrogenic properties and induces hormonal regulation in rats with PCOS. RSV treatment generates enhanced levels of plasma adiponectin and estradiol, recuperation of normal ovarian morphology, and elevated ovarian expression of nesfatin-1 and aromatase at the RNA and protein levels [163].

The nutraceutical preparation mixture consisting of RSV, alpha-lipoic acid, folic acid, vitamin D, and vitamin B provided favorable results for alleviating the metabolic profile of women with PCOS. Following 12 weeks of therapy, body mass index, anthropometry, and bioimpedance hallmarks were all decreased in the treated patients [164].

In a study involving 110 patients, the treatment with both RSV and myoinositol was substantially more efficient than metformin and pioglitazone in improving abnormal endocrine, metabolic markers and stress burden and has excellent potential in the high-risk category of obese, oligo-anovulatory women with PCOS [165]. Another inquiry established that the protective actions of RSV on ovulation in rats with PCOS are correlated to the modulation of glycolysis-dependent mediators, such as pyruvate kinase isozyme M2, lactate dehydrogenase A, and SIRT1 [166].

### 5.6. Anti-Aging Properties

RSV and caloric restriction are very potent treatment alternatives for anti-aging and were compared in vitro and in vivo regarding their activities in longevity-related gene silencing information regulator SIRT1. Interestingly, 10 µM RSV in vitro and a higher dose in vivo exhibited superior anti-aging effects and SIRT1-stimulation degrees compared to caloric restriction, implying the promising potential of the phytocompound as a caloric restriction-mimetic [167].

RSV has a strong beneficial effect on lesion healing, regeneration, and photoaging of the epidermis. Through the medium of complex mechanisms and pathways, it offers protection against UVB radiation, which is the principal element in skin aging development. Moreover, it increases collagen synthesis by stimulating the ERs and alleviates wrinkles. In deteriorated tissues, it expedites epidermis regeneration and healing by triggering VEGF. Undeniably, RSV is a valuable polyphenol in cosmetology, dermatology, and plastic surgery, as it is utilized in anti-aging products or administered topically on scars and lesions. Future approaches may involve RSV in the pharmacotherapy of various dermatoses [168].

## 6. Hepatoprotective Effect

### 6.1. Liver Ischemia–Reperfusion

Ischemia–reperfusion (I–R) of the liver is a complex process that occurs during transplantation and surgery. The destructive repercussions of I–R derive from the intense generation of ROS, which can provoke instantaneous tissue deterioration and numerous disastrous cellular reactions, including apoptosis, organ collapse, and inflammation. The formation of ROS in the I–R condition can impair the antioxidant system and produce liver depreciation. RSV has been demonstrated to possess antioxidant effects on liver lesions caused by I–R in mice. The results revealed that hepatic ischemia happens after liver transplantation and that the administration of RSV in advance of reperfusion improves graft survival as a consequence of the suppression of protein response signaling pathways, particularly protein kinase R-like ER kinase (PERK) and inositol-requiring enzyme 1alpha (IRE1$\alpha$) [169].

### 6.2. Non-Alcoholic Fatty Liver Disease

NAFLD is a progressive disorder concerning the liver that compels efficient therapies to prevent liver damage. A micronized formulation of *t*RSV, with higher absorption and enhanced antioxidant action, proved to be more efficient than simple RSV preparations in

humans. It was successful in decreasing the liver adipose tissue, including reducing insulin resistance and hepatic enzymes, such as serum glutamate pyruvic transaminase and gamma-glutamyl transpeptidase [170].

NAFLD, a common cause of chronic liver disease worldwide, is linked to increased body weight and obesity. It leads to fat accumulation in the liver, resulting in insulin resistance, inflammation, oxidative stress, and altered lipid and glucose profiles. RSV may help manage NAFLD due to its anti-inflammatory and antioxidant properties and effects similar to calorie restriction. In vitro and in vivo models show that RSV reduces liver fat and improves lipid and glucose metabolism through the activation of AMPK, SIRT1, and NF-κB signaling pathways [171].

Polydatin is a natural precursor and glycosylated type of RSV with different bioactivity. The responses of polydatin and RSV were measured against energy homeostasis disproportion in mice with high-fructose nutrition. Over-nourishment-activated ROS stimulate AMPK inhibition and metabolic dysfunction. It was firstly validated that polydatin treatment enhances the fecal concentrations of valeric and caproic acids by altering gut microbiota; therefore, stimulating AMPK activation might be the intrinsic mechanism that polydatin is preferable to RSV in ameliorating lipid dysmetabolism [172].

### 6.3. Anti-Hepatotoxic Activity

The liver is a crucial organ in the body and operates essential functions. RSV has favorable pharmacological effects in the therapy of several liver pathologies, such as fatty hepatitis, liver steatosis, liver carcinoma, and liver fibrosis, through its anti-inflammatory, anti-apoptotic, antioxidative, and hepatoprotective actions. RSV reduces liver inflammation by inhibiting numerous proinflammatory cytokines. Additionally, it suppresses the NF-κB transcription factor, which initiates the inflammatory pathway and deactivates the PI3K/Akt/mTOR axis to inflict apoptosis. Furthermore, it decreases oxidative stress in hepatocytes by substantially diminishing MDA and nitric oxide (NO) and enhancing CAT and SOD. Because of its antioxidant, anti-inflammatory, and anti-fibrotic effects, RSV curtails liver injury indicators. It is also a reliable natural antioxidant that improves the hepatotoxicity of noxious chemicals [173].

The hepatoprotective effect of RSV against ethanol-induced oxidative stress was examined in vivo. The fundamental mechanisms by which RSV exhibits its antioxidative activity on hepatic cells are explained through the generation of SOD activity and gene expression [174].

Recent investigations targeted the protective actions of RSV and avocado oil against paracetamol-generated hepatotoxicity in rats. The outcome illustrated that the total oxidant status was enlarged in the paracetamol group, while the total antioxidant status was enhanced in the RSV and RSV plus avocado oil groups. The histopathological analysis determined necrotic regions in the livers of the rats. The administration of both RSV and avocado oil in advance inverted the oxidative stress parameters, DNA deterioration, and necrosis caused by paracetamol [175].

## 7. Cardioprotective Properties

### 7.1. Coronary Heart Disease

RW is the most beneficial alcoholic beverage in reducing the incidence of coronary heart disease (CHD). There have been investigations of the mechanism of *t*RSV on human platelet aggregation and synthesis of eicosanoids from arachidonate by platelets. Numerous wine phenolic compounds and antioxidants were compared, and *t*RSV and quercetin showed a dose-related reduction in platelet aggregation caused by thrombin and adenosine diphosphate (ADP). Moreover, *t*RSV suppressed the formation of thromboxane B2, hydroxyheptadecatrienoate and 12-hydroxyeicosatetraenoate, from arachidonate, conditioned by the dosage. This information emphasizes the protective effect of RW against atherosclerosis and CHD due to tRSV [176].

### 7.2. Myocardial Ischemia–Reperfusion

The pre-administration of RSV has a beneficial influence on mitochondrial activity and myocardial I–R damage due to the suppression of stromal interaction molecule 2 (STIM2) through the medium of miRNA-20b-5p. These data have been recently verified in a study on 90 rats, involving assays to identify the expressions of Bcl-2, Bcl-2-associated X protein, and cleaved-cysteine proteinase 3, together with the levels of mitochondrial membrane potential, ROS, and adenosine triphosphate (ATP) [177].

### 7.3. RSV as Vascular Protective Natural Compound

Vascular metabolic dysfunction is exhibited in numerous recurrent disorders, such as atherosclerosis, hypertension, and diabetes mellitus. Current appraisals have determined that RSV has a valuable influence on repairing metabolism in endothelial cells and vascular smooth muscle cells. The regulation of cell metabolism activity is connected to the suppression of glucose uptake, inhibition of glycolysis, prevention against damages caused by fatty acids through stimulation of fatty acid oxidation, reduction in lipogenesis by increasing lipolysis, and elevation of glutamine uptake and synthesis [178]. Vascular smooth muscle cell senescence is a critical element that stimulates the progression of CVDs. RSV has the ability to prevent vascular senescence via inhibiting oxidative stress and mitochondrial alteration by activating E2F1/SOD2 pathway [179].

### 7.4. RSV and Calcium Fructoborate Supplementation against Stable Angina Pectoris

The clinical and biological conditions of humans diagnosed with stable angina pectoris were investigated in a study that proposed to assess the results of per os treatment with calcium fructoborate (CaFB) and RSV. The following observations were highlighted: hs-CRP, an inflammation biomarker, was seriously reduced; the N-terminal prohormone of brain natriuretic peptide, a left ventricular function marker, was substantially decreased; the lipid markers were also influenced. The findings determined that the association of RSV with CaFB has favorable consequences in patients diagnosed with angina pectoris, improving their quality of life [180].

## 8. Immunostimulatory Activity

RSV interacts with immune cells and modulates the innate and adaptive immunity and the activation of macrophages, T-cells, NK cells, and CD4+ CD25+ T-cells. RSV enters the cell by binding to specific receptors (e.g., integrin receptor $\alpha v\beta 3$) besides the well-known mechanisms of passive diffusion and mediated endocytosis. By activating SIRT1, RSV decreases the NF-κB-induced expression of COX-2, MMP-1 and MMP-3, TNF-$\alpha$, IL-1$\beta$, and IL-6, with beneficial effects on some autoimmune diseases, such as rheumatoid arthritis, systemic lupus erythematosus, encephalomyelitis, psoriasis, T1DM, inflammatory bowel disease [181–183].

The in vitro effects of RSV on cytokine biosynthesis, inflammatory genes, and cell survival and proliferation were investigated in a human monocytic cell line (THP-1) and phorbol 12-myristate 13-acetate differentiated human THP-1-derived macrophages. Using physiological concentrations, RSV caused S phase arrest and inhibited the proliferation of THP-1 monocytes. By comparison, at pharmacological doses, RSV leads to G0/G1 phase arrest and induces cell apoptosis. Moreover, RSV promoted an inflammatory state in THP-1 monocytes and induced anti-inflammatory properties in THP-1-derived macrophages [184].

In an experimental model, RSV promoted recovery of immune function by activating the JNK/NF-κB pathway in splenic lymphocytes of immunosuppressive mice. The study demonstrated that RSV could modulate JNK, c-jun, NF-κB, and IκB kinase in vivo expressions [185].

Another in vivo study that highlighted the immunostimulatory activity of RSV was performed on Wistar rats irradiated with 2 Gy single doses. After irradiation, RSV 10 or

100 mg/kg was intraperitoneally administered for 30 days. The results highlighted that RSV 100 mg/kg modulated the liver inflammatory profile of immune response and circulating endothelial cells (CECs) population during acute whole-body gamma irradiation [186].

The obstructive effects of RSV on the release of mediators from bone marrow-derived mouse mast cells were verified in vitro. Regulated by immunoglobulin E (IgE), the discharge of histamine, TNF-$\alpha$, leukotrienes, and PGD2 were seriously suppressed by RSV; in addition, A23187-modulated release of histamine and leukotrienes was also actively decreased. RSV did not exhibit cytotoxic effects against mast cells and can, therefore, be an important non-selective blocker of mediator release from activated mast cells [187].

The study of the influence of RSV alone and combined with cyclosporin A on the propagation of human peripheral blood T-lymphocytes, conversion into lymphoblasts, as well as IL-2 and INF-$\gamma$ generation, revealed that RSV can significantly inhibit the proliferation and transformation of lymphocytes and the synergic mixture of RSV and cyclosporine A can stimulate immune suppression [188].

### 9. Antioxidant Effect

RSV is a well-known natural compound active as a scavenger of free radicals. In several in vitro antioxidant assays, such as 2,2′-azino-*bis*(3-ethylbenzothiazoline)-6-sulfonic acid (ABTS), 2,2-diphenyl-1-picrylhydrazyl (DPPH), *N,N*-dimethyl-*p*-phenylenediamine (DMPD), $H_2O_2$, NO, oxygen radical absorbance capacity (ORAC), total antioxidant activity (TAC), cupric reducing antioxidant capacity (CUPRAC), ferric reducing antioxidant power (FRAP), RSV exhibited radical scavenging, reducing power, $Cu^{2+}$ and $Fe^{2+}$ chelating effects. Also, RSV induces an inhibition of lipid peroxidation of linoleic acid emulsion greater than $\alpha$-tocopherol, 6-hydroxy-2,5,7,8-tetramethylchroman-2-carboxylic acid (Trolox), butylated hydroxytoluene (BHT), and butylated hydroxyanisole (BHA) [189–191].

RSV has in vitro and in vivo antioxidant effects, both directly and indirectly, by the inhibition of ROS production and modulation of several antioxidant enzymatic systems, respectively [192,193].

Nanoencapsulation using soy lecithin/sugar esters, Tween 20/glycerol monooleate, Tween 80/dodecanol (Dod), and Span 80/Dod preserved physicochemical stability and in vitro antioxidant capacity of *t*RSV as shown in chemical (DPPH, ABTS) and cellular assays (Caco-2 line), comparing with dimethyl sulfoxide (DMSO)-solubilized unencapsulated *t*RSV [194,195].

RSV is a powerful natural antioxidant and an extremely photosensitive compound. Moreover, after oral administration, RSV has low bioavailability due to its low solubility and rapid first-pass effect [196,197]. In this regard, the chemical stability, oral bioavailability, and antioxidant potential of *t*RSV were improved using both lipid-coated nanocrystals [198] and an electrospun nanofiber system based on polyvinylpyrrolidone (PVP) and hydroxypropyl-β-cyclodextrin (HPβCD) [199].

RSV has been determined to defend LDLs against peroxidative degeneration. Being a tyrosine kinase suppressor, it also has the capacity to inhibit intracellular adhesion molecule 1 and vascular cell adhesion molecule 1 expression and to stop the attachment of monocytes and granulocytes to endothelial cells. This mechanism is autonomous from its antioxidant action and provides a different interpretation [200].

Nutritional addition of RSV inhibits lipid peroxidation, the critical consequence of oxidative stress caused by chronic ethanol treatment. During a study on ethanol-treated rats, RSV substantially counteracted the elevation in MDA, a marker of oxidative damage, concentrations in the liver, heart, brain, and testes, demonstrating its protective effect [201]. Numerous assays evaluated the inhibitory effect of RSV in oxidative stress. However, the association of RSV and vitamins C and E is more efficient in protecting the cells than any of these antioxidants alone [202].

In a randomized clinical trial on older adults with T2D, RSV treatment, 1000 mg/day for six months, exhibited a statistically significant antioxidant effect highlighted by the decreasing of oxidative stress biomarkers and increasing of TAC and SIRT1 levels [203].

## 10. Bone Protection

Several experimental models in senile, ovariectomized, postmenopausal, and osteoporotic rats highlighted the cellular and molecular mechanisms of RSV as a bone-protective agent, e.g., inhibition of oxidative stress, inflammation, and NF-κB/receptor activator of NF-κB ligand (RANKL)-mediated osteoclastogenesis; activation of SIRT1; stimulation of osteogenesis and mesenchymal stem cells differentiation to osteoblasts; and modulation of bone morphogenetic protein 2 (BMP2), FoxOs, MAPK/JNK/ERK, miRNAs, PI3K/AKT, and Wnt/β-catenin signaling pathways [204–206]. Also, in a rat model of postmenopausal osteoporosis, RSV supplementation, 10, 20, or 40 mg/kg/day for eight weeks, promoted osteoblastic differentiation and inhibited osteoclastic differentiation by regulating autophagy [207].

Nutritional supplementation of RSV has beneficial effects against hip fracture risk in elderly patients. This information was documented by research involving recently diagnosed patients with hip fractures. The investigations included the administration of RSV and piceid, in addition to RSV-rich foods, such as grapes, apples, and nuts. The results were more conspicuous in female and non-obese subjects and highlighted that alimentation with RSV correlates with decreased risk of hip fracture [208].

In a randomized, double-blind, placebo-controlled trial in postmenopausal women, RSV supplementation, 75 mg twice daily for 24 months, increased bone mineral density (BMD) in the femoral neck and lumbar spine. Also, the bone-protective potential of RSV treatment was strengthened by co-administration of calcium and vitamin D [209]. RSV supplementation, 250 mg twice daily for six months, in a randomized, double-blind, placebo-controlled trial, prevented the loss of BMD in T2DM patients, who are known to have an increased risk of fractures [210]. Moreover, in middle-aged obese men with MetS, RSV 1000 mg oral treatment daily for 16 weeks prevented bone loss by increasing lumbar spine trabecular volumetric BMD and bone alkaline phosphatase level, compared with placebo [211].

## 11. Wound Healing Properties

Compared with the oral route of administration, when applied topically, RSV has a good skin penetrating capacity and a low rate of degradation, ensuring an improved and long-lasting effect [212].

RSV supplementation has beneficial effects on wound healing because it interferes with cellular and molecular pathways responsible for pathophysiological processes such as oxidative stress, inflammation, infection, autophagy, collagen proliferation, and angiogenesis [213,214].

In vitro, RSV accelerated cell proliferation and migration by the regulation of miR-212/caspase-8 axis in lipopolysaccharide (LPS)-treated human epidermal keratinocyte (HaCaT) cell culture [215] and by the up-regulation of nuclear Nrf2 (N-Nrf2) and antioxidant manganese SOD (MnSOD) in human umbilical vein endothelial cells (HUVECs) [216].

In several experimental murine models, RSV treatment increased cutaneous wound healing, inhibited excessive scarring, prevented skin photoaging, and stimulated chronic wound healing [215–218]. In STZ-induced C57/B6 diabetic mice, RSV 10 μmol/L, injected around the injury, induced M2 macrophage polarization, decreased the level of pro-inflammatory cytokines (TNF-α, IL-6, IL-1β), and accelerated the skin wound healing [219].

The role of local and systemic injections of RSV on open cutaneous wound healing was compared in rats. Both of them elevated the histological scores for collagen deposition, chronic inflammation, and granulation. Neovascularization rates were substantially higher in the subcutaneous RSV treatment category than in the intraperitoneal one. Both

systemic and local administrations importantly boosted wound healing and enhanced the tensile strength of the epidermis. Local subcutaneous injection of RSV emerged as a better therapeutic approach than systemic treatment due to the neovascularization to help wound healing [220]. On the other hand, topical administration of 5% RSV ointment was demonstrated to stimulate burn wound healing by enhancing the progress of wound contraction via collagen fiber synthesis, granulation tissue creation, and epithelial regeneration in rats [221].

Diabetic injuries are problematic to heal because of constant inflammation and reduced angiogenesis. Platelet-derived extracellular vesicles are abundant in growth factors and cytokines that stimulate proliferation and angiogenesis. Nevertheless, individual medication therapy has reduced potency and release ability that can be improved by bioengineering, which combines active substances and materials to obtain synergic medication. A composite hydrogel was used as a wound cover for continuous RSV delivery from mesoporous silica NPs, combined with platelet-derived extracellular vesicles, and showed promising effects in modulating inflammation and angiogenesis of diabetic injuries and accelerating wound restoration [222,223].

## 12. Anti-Inflammatory Effect

At the cellular and molecular level, RSV has a regulatory effect on the inflammatory response due to the influence it exerts on various signaling pathways that protect the body also against oxidative stress, such as AA, NF-κB, MAPK, SIRT1/AMPK, JNK, β-catenin, glycogen synthase kinase-3 beta (GSK-3β), intercellular adhesion molecule-1 (ICAM-1), monocyte chemoattractant protein-1 (MCP-1), AP-1, Nrf2, and heme oxygenase-1 (HO-1) [224–226].

A recent study highlighted the in vitro anti-inflammatory effect of RSV-enriched Dongjin rice (DJ526) callus extract on LPS-stimulated RAW264.7 macrophages. RSV treatment significantly reduced the levels of pro-inflammatory factors, such as NO, PGE2, COX-2, iNOS, TNF-α, IL-6, and IL-1β [227].

RSV protects the gut barrier and regulates the gut microbiome, acting for the prevention of intestinal inflammation [228]. In this respect, RSV treatment mitigates the intestinal mucosal damage, oxidative stress, and inflammation caused by zearalenone (ZEA) mycotoxin exposure in mice by modulating Nrf2/HO-1 and NF-κB signaling pathways [229].

In an experimental model of monocular form deprivation (MFD)-induced myopia, RSV supplementation exhibited an anti-inflammatory effect by decreasing the expression levels of TNF-α, IL-6, IL-1β, TGF-β, MMP-2, and NF-κB and increasing the level of collagen I [230].

Oxaliplatin is an average chemotherapy medication that causes neurotoxicity. To examine possible treatment variants for the neuropathic pain and inflammatory reaction induced by oxaliplatin, RSV was intrathecally injected into the spinal cord of rats. The findings revealed that RSV administration decreased COX-2 expression and obstructed ROS generation. The anti-inflammatory mechanism of RSV inhibited astrocytic activation and induced an antinociception effect that relieved neuropathic pain [231].

Assays of the significance of RSV in diabetic peripheral neuropathy emphasized the following results: the discomfort and temperature responses of diabetic mice were ameliorated; Nrf2 level was enhanced in the diabetic peripheral nerves, and NF-κB axis suppression preserved nerves. RSV regulates the anti-inflammatory microenvironment of peripheral nerves by stimulating Nrf2 activation and the expression of phospho-p65 [232].

Due to its reduced bioavailability, the RSV NPs are a better solution and have a higher neuroprotective role in a study involving rats with middle cerebral artery occlusion. Animals were tested for infarct volume and oxidative, inflammatory, and apoptotic markers. The mechanism of RSV involved the increased expressions of caspase-3 and caspase-9, and IL-1, IL-6, and TNF-α cytokines. Therefore, these findings show that suppression by RSV NPs has a therapeutic effect on ischemic stroke [233].

Recent explorations presented that the treatment with either monomeric tRSV or RSV dimer—gnetin C from *Gnetum gnemon* L., gnetum (*Gnetaceae*) seed extract—limits periodontitis, with higher decrease in bone loss being exhibited in the dimer group than the monomer batch and that these effects are connected to reduced oxidative stress and consequently minimization of local inflammation by downregulation of IL-1β, a proinflammatory cytokine [234]. Another finding suggests that RSV suppresses systemic local inflammatory markers and systemic endotoxin and implies that 500 mg per day of RSV is the optimal dosage for patients with periodontitis [235].

## 13. Antimicrobial Activity

RSV manifests antimicrobial action opposite to a remarkably extensive spectrum of bacterial and fungal species. It has the ability to modify the bacterial expression of a virulence nature, affecting the cell cycle, promoting decreased toxin formation, suppressing biofilm generation, limited motility, and influencing the quorum sensing (QS) for both health benefits and as a natural food preservative [236]. In association with regular antibiotics, RSV stimulates the activity of aminoglycosides against *Staphylococcus aureus* while it inhibits the noxious activity of fluoroquinolones against *S. aureus* and *Escherichia coli*. Moreover, topical administration of this nutraceutical has proven beneficial in acne lesions provoked by *Propionibacterium acnes*. Recent data illustrate that the synergic combination of RSV with specific antibiotics may increase their antimicrobial potency, alleviating the crescent issue of antimicrobial resistance [237,238].

The virulence factors of methicillin-resistant *S. aureus* (MRSA) clinical isolates, such as hemolysin, hemagglutination, protease, biofilm, and lecithinase, were significantly inhibited by tRSV 50 μg/mL and curcumin 20 μg/mL in vitro separate administration. Consequently, tRSV and curcumin could be used as an alternative therapy against bacterial resistance to antibiotics [239].

RSV inhibits *E. coli* virulence factors and cell growth by increasing DNA fragmentation, blocking Z-ring formation, and ftsZ gene expression. In addition, RSV administration increased cell elongation and ROS production in *E. coli* [240].

RSV and polymyxin B act synergistically against multidrug-resistant (MDR) Gram-negative *Klebsiella pneumoniae* and *E. coli* bacterial strains. Minimum inhibitory concentrations (MICs) of polymyxin B were significantly lowered by RSV (32 μg/mL to 128 μg/mL, optimal concentrations depending on bacteria). The study evidenced that the sensitivity of MDR Gram-negative bacterial strains to polymyxin B can be increased by RSV treatment [241].

Topical bacterial infection produced by MDR *Pseudomonas aeruginosa* is very difficult to treat, using polymyxins as the last alternative therapy, with unfavorable pharmacodynamics and PK, improbable to attain effective blood levels. Well-tolerated RSV had a synergic effect combined with polymyxin B therapy when analyzed in vitro on antibacterial and anti-biofilm activities [242]. It was also established that Chilean RWs have an antibacterial effect against *Helicobacter pylori* due to the existence of RSV in their composition [243,244].

A recent in vitro study highlighted that RSV, chlorhexidine digluconate (CHX), and benzalkonium chloride (BZK) co-administration could be successfully applied against nosocomial pathogens responsible for healthcare-associated infections, such as *Burkholderia* spp., *K. pneumoniae*, *P. aeruginosa*, *Stenotrophomonas maltophilia*, and *Candida albicans* clinical isolates [245].

RSV treatment (50 and 100 μg/mL) can also inhibit motility, QS system, and biofilm formation of *Aeromonas hydrophila*, a Gram-negative aquatic bacterium responsible for septicemia in humans [246].

*C. albicans* is the principal candidiasis-inducing fungal pathogen in humans, and one of its most critical virulence elements is its capacity to synthesize biofilms. The antifungal effects of RSV (10–20 μg/mL) alter the morphological transition of *C. albicans* following several hypha-inducing circumstances and reduce the growth of the yeast form and

mycelia. Therefore, RSV has promising anti-*Candida* actions via suppressing existing or in the formation process *C. albicans* biofilms [247,248].

## 14. Antiviral Properties

Several studies found that RSV has a cooperative effect in association with nucleoside analogs, such as zidovudine, zalcitabine, and didanosine, stimulating their activity against human immunodeficiency virus type 1 (HIV-1). Individual RSV did not produce cell toxicity and inhibited viral replication, while in peripheral blood mononuclear cells infected with human T-cell lymphotropic virus-IIIB (HTLV-IIIB) isolate of HIV, it decreased the inhibitory concentration of the nucleoside analogs. An identical antiviral effect was documented when didanosine was merged with RSV in peripheral blood mononuclear cells infected with HIV-1. The inclusion of RSV determined a 10-time amplification of the antiviral action of didanosine in infected monocyte-derived macrophages. In a test on HTLV-IIIB-infected T-lymphocytes, the RSV–didanosine mixture, but not independently, inhibited the viral infection [249].

RSV has been demonstrated to have an anti-herpes simplex virus (HSV) effect in vitro. To examine its activity in vivo, the abraded skin of mice was infected with HSV-1, and RSV cream was applied locally. The following findings were made: 25% RSV cream successfully inhibited lesion progression, while 12.5% RSV cream had the same effect only if applied one hour after infection; neither one was effective if applied twelve hours after infection. In addition, RSV cream, 10% docosanol cream, and 5% acyclovir ointment were compared. RSV and acyclovir substantially obstructed the progression of HSV-1-caused epidermis lesions, while docosanol had no influence. The application of RSV provoked no dermal toxicity, such as erythema, scaling, crusting, lichenification, or excoriation [250].

Regarding human cytomegalovirus replication, extremely high concentrations of RSV were necessary to generate cytotoxicity as opposed to developing or stagnant human embryonic lung fibroblasts. RSV obstructed the virus-activated epidermal growth factor receptor and PI3K signal transduction, as well as NF-κB and Sp1 transcription factor activation, immediately post-infection. The mechanisms of the cytomegalovirus, such as DNA replication, PI3K signaling, and transcription factor activation, were all suppressed by RSV. However, the virucidal action was incompatible, and the antiviral effect of RSV was significantly reduced when administered four hours after infection. As a result, this phytocompound engaged amid attachment and entry via arrest of epidermal growth factor receptor activation and its succeeding effectors [251].

Worldwide, rotavirus is the principal originator of viral gastroenteritis in children, with no efficient therapy. The anti-rotavirus impact of RSV was inspected in vitro, including the production of virion descendants, viral polyprotein expression, genomic RNA synthesis, antigen clearance, and alterations in proinflammatory cytokines/chemokines in infant mouse cells. The assays concluded that RSV strongly limited rotavirus replication by inhibiting RNA formation, protein expression, viroplasm plaque generation, progeny virion synthesis, and cytopathy. RSV treatment ameliorated the gravity of diarrhea, reduced viral titers, and alleviated correlated symptoms [252].

The high-occurrence hepatitis B virus infection requires successful therapeutics. A recent study prospected the influence of RSV on hepatitis B virus replication, utilizing in vitro and in vivo analysis, and confirmed that it suppresses cytotoxicity and virus replication. Through its mechanism of action via reducing miR-155 expression and stimulating autophagy, RSV proved to be a promising novel approach for the treatment of hepatitis B virus infection [253].

## 15. Risks and Interactions of Resveratrol

RSV is widely acknowledged for its chemopreventive and antioxidant properties, yet recent studies underscore its potential as a pro-oxidant agent, capable of paradoxically contributing to disease pathology. Although renowned for its ROS-scavenging activity, RSV can also generate ROS, leading to oxidative stress. This biphasic nature manifests in

distinct ways depending on dosage and exposure time. At low concentrations (0.1–1.0 µg/mL), it enhances cell proliferation, whereas at higher concentrations (10.0–100.0 µg/mL), it induces apoptosis and reduces mitotic activity in tumor and endothelial cells. Such dose-dependent effects are further influenced by the time of administration, as it behaves as an antioxidant during the dark phase but switches to a pro-oxidant during light periods, indicating a complex interplay between RSV's pro- and antioxidant activities. Additionally, RSV's regulation of mitochondrial function can induce oxidative stress in aged organisms due to a compromised antioxidant defense system, thereby increasing the production of superoxide radicals. Furthermore, its phytoestrogenic properties allow it to act as an agonist or antagonist to ERs, potentially disrupting hormone signaling pathways in certain cancer cell lines. Notably, in aging mice on high-protein diets, RSV was found to exacerbate cardiovascular risk factors by promoting inflammation and superoxide production while decreasing aortic flexibility. These findings emphasize that while RSV offers health benefits, its dose-dependent, dualistic nature necessitates careful consideration to mitigate potential adverse effects [254,255].

RSV, a promising compound due to its generally non-toxic profile, requires careful dosing to maximize benefits while minimizing adverse effects. While inducing apoptosis in tumor tissues with relatively minimal impact on adjacent healthy cells, higher doses exceeding 2.5 g/day can cause gastrointestinal disturbances like nausea, diarrhea, and vomiting, as well as liver dysfunction, particularly in patients with NAFLD. Notably, RSV remains safe at doses up to 5 g/day over multiple days in healthy individuals. However, challenges arise in understanding its effects due to the metabolism of RSV by gut microbiota [254,255].

Studies involving animal models highlight the risks of RSV at very high doses. In hypercholesterolemic rabbits, RSV (1 mg/kg) surprisingly promoted atherosclerosis rather than protecting against it. In rats, high doses (3000 mg/kg) led to nephrotoxicity, resulting in elevated blood urea nitrogen and creatinine levels, renal lesions, and renal tubule dilation. High doses also resulted in anemia, weight loss, and elevated liver enzymes, indicating potential liver toxicity [254].

RSV has biphasic effects, meaning that while it can inhibit tumor growth, it also reduces cell growth and induces apoptosis in normal cells at higher concentrations. Doses of 1800 mg/kg reduced the lifespan of mice to 3–4 months. Although it is generally well-tolerated, diarrhea was frequently reported in PK studies with 2000 mg doses administered twice daily. This emphasizes the importance of identifying optimal dosing and administration routes for safe and effective use [254].

RSV's interactions with various medications, particularly through modulation of cytochrome P450 (CYP) enzymes and transporters, highlight the importance of caution in its supplemental use. It inhibits CYP3A4, potentially reducing the metabolic clearance of drugs that rely on this enzyme, leading to elevated bioavailability and increased toxicity. Patients on drugs like tamoxifen, which depend on CYP enzymes for efficacy, could be significantly impacted by high doses of RSV [254,255].

Moreover, RSV interferes with key transport proteins, including P-glycoprotein (P-gp), multidrug resistance-associated protein 2 (MRP2), and organic anion transporter 1/3 (OAT1/OAT3). Although these effects are still not fully understood, higher doses may compete with other polyphenols for transport, diminishing their absorption and therapeutic effects. Additionally, the absorption, distribution, and elimination of RSV in humans need further exploration for accurate prediction of drug interactions [254,255].

Furthermore, RSV inhibits human platelet aggregation in vitro, suggesting that high doses could increase the risk of bruising and bleeding, particularly when combined with anticoagulant, antiplatelet, or non-steroidal anti-inflammatory drugs (NSAIDs). Thus, further research into its drug interaction mechanisms is warranted to ensure safe and effective use [254,255].

### 16. Pharmacokinetics of Resveratrol: Challenges in Absorption, Metabolism, and Therapeutic Efficacy

Despite being well-absorbed orally, RSV presents significant PK challenges that hinder its clinical utility in chemotherapeutics. Its oral bioavailability is only about 70%, yet its systemic availability is reduced to a mere 5% due to extensive hepatic and intestinal metabolism via glucuronidation and sulfation, leading to less active metabolites. Additionally, oxidative enzymes, air oxidation, light exposure, and high temperatures can induce *trans*-to-*cis* conversion, further diminishing its efficacy. Compounding these challenges is its poor water solubility of approximately 0.03 mg/mL, which restricts its absorption and systemic circulation [256].

Recent research has focused on new delivery systems to address these PK barriers. Nanoformulations show promise in enhancing solubility, preventing degradation, and controlling drug release kinetics. Other strategies, including metal-based carriers, derivative compounds, and polyherbal formulations, are also being explored [257].

Despite these efforts, evidence shows that RSV is quickly metabolized and excreted as sulfate and glucuronide conjugates. Studies involving human urine and blood samples consistently reveal low levels of unmetabolized RSV and high levels of its conjugated metabolites, indicating rapid metabolism and extensive enterohepatic recirculation. For instance, a study on oral RSV administration found that only 2% of the dose was excreted unchanged, confirming an oral bioavailability as low as 5% [258].

Moreover, its bioavailability varies among individuals, influenced by factors like diet composition and metabolism. While complex meals may improve absorption, obtaining the necessary therapeutic doses remains challenging. The variability of RSV's bioavailability, particularly in cancer patients who may experience altered metabolism, suggests a significant gap in understanding its therapeutic efficacy and warrants further investigation [257].

### 17. Strategies for Enhancing Resveratrol's Bioavailability

The druggability of RSV, a compound known for its health benefits, can be significantly improved through various strategies. One key approach is co-administration with piperine, an alkaloid derived from *Piper nigrum* L., black pepper (*Piperaceae*). Piperine inhibits the glucuronidation of RSV, a metabolic process that rapidly deactivates the compound in the liver and intestines. By co-administering piperine, the maximum serum concentration ($C_{max}$) of RSV is increased by over 15 times, while the total systemic exposure, measured as the area under the curve (AUC), more than doubles. This strategy considerably reduces the rapid metabolism and elimination that typically limit RSV's effectiveness [259].

Another method involves developing RSV analogs with improved PK and therapeutic properties. Modifying the chemical structure has led to promising analogs, such as tetramethoxystilbene, isorhapontigenin, and RSV trimethyl ether, which exhibit greater bioactivity and stability. These analogs display superior antitumor potential and could serve as alternatives in future clinical development [260].

Furthermore, utilizing nanoformulations and advanced delivery systems enhances RSV's stability and tissue targeting. Lipid NPs, micelles, and cyclodextrin-based carriers protect the compound from degradation while improving tissue-specific delivery. Intratracheal administration of lipid NPs ensures localized, sustained release for pulmonary treatments. Additionally, novel administration routes, like transdermal, buccal, and direct nose-to-brain, have shown promise in enhancing absorption and targeted delivery [257].

Lastly, prodrugs such as tri-O-acetyl-RSV transform RSV into more bioavailable forms, increasing the plasma concentrations of active *t*RES. Together, these strategies effectively improve the PK properties and therapeutic potential of RSV, thus enhancing its overall druggability [261].

## 18. UHPLC–MS Analysis of *t*RSV in Romanian Wines

### 18.1. Chemicals and Reagents

For the analysis of *t*RSV in wine samples through UHPLC–MS, the following chemicals and reagents were used: *t*RSV was purchased from Sigma-Aldrich (Taufkirchen, Germany) and utilized as a reference for identification and quantification processes. Acetonitrile, obtained from Merck (Darmstadt, Germany), served as one of the mobile phase components due to its excellent solvent properties and compatibility with MS. Formic acid, also sourced from Merck, was employed as a mobile phase modifier to enhance the ionization of *t*RSV during the MS detection phase. Ultrapure water was produced locally in our laboratory using the HALIOS 6 (Neptec, Montabaur, Germany) ultrapure water system and was utilized as the aqueous component of the mobile phase and for all preparations requiring solvent. All reagents were of analytical grade and were used without further purification to ensure the integrity and reproducibility of the results, ensuring optimal separation, identification, and quantification of *t*RSV in the studied wine samples.

### 18.2. Calibration Curve for tRSV Quantification

A stock solution of *t*RSV was prepared at a concentration of 1 mg/mL. From this stock solution, serial dilutions were made to create calibration standards at the following concentrations: 244.141, 122.070, 61.035, 30.518, and 15.259 ng/mL. These standards were prepared by diluting the stock solution with the initial mobile phase to achieve the precise concentrations needed. These diluted standards were then analyzed using the UHPLC–MS system under identical conditions to those of the wine samples. The resulting peak areas for *t*RSV at *m/z* 227 were plotted against their respective concentrations to construct the calibration curve. The linearity of the calibration curve was validated by the correlation coefficient ($R^2$) obtained from the linear regression of the data. The curve was crucial for quantifying the concentration of *t*RSV in the wine samples by interpolating the sample peak areas against the calibration data. Measurements were made in triplicate to ensure the accuracy and reliability of the data. The limits of detection (LOD) and quantification (LOQ) for *t*RSV were determined based on the standard deviation (SD) of the response and the slope of the calibration curve, ensuring precise and reliable quantification [262–264].

### 18.3. Wine Samples

The study focused on the analysis of *t*RSV in four wine samples, all sourced from local Romanian vineyards to represent both white and red varieties, providing a comprehensive overview of *t*RSV content across different types of wine. The white wine samples included a Fetească Regală (W1), a traditional Romanian grape variety known for its aromatic profile and high acidity, and a Dry Muscat (W3), noted for its sweet floral aromas and dry finish. The RW samples consisted of a Fetească Neagră (W2), an indigenous Romanian variety valued for its deep color, medium tannins, and berry flavors, and a Cabernet Sauvignon (W4), a globally recognized grape chosen for its robust structure and potential for aging. Each wine sample was stored and handled under controlled conditions to preserve its chemical integrity until analysis, aiming to explore the influence of grape type and wine style on the levels of *t*RSV and enhance our understanding of how this phenolic compound varies within Romanian wines [265].

### 18.4. UHPLC–MS Analysis

The UHPLC–MS analysis of *t*RSV in wine samples was conducted using a Waters Arc System coupled with a Waters QDa detector. Initially, all samples were diluted five-fold and then filtered through 0.2 μm filters before being injected into the system, with an injection volume set at 5 μL. The mobile phase comprised two components: (A) was 0.1% formic acid in water, and (B) was 0.1% formic acid in acetonitrile. The flow rate was maintained at 0.8 mL/min. The gradient program was as follows: from 0 to 1.8 min, (B) increased from 2% to 9%; from 1.8 to 4 min, (B) held steady at 9%; from 4 to 10 min, (B) rose

from 9% to 30%; from 10 to 15 min, (B) escalated from 30% to 90% and was maintained at 90% until 16 min; from 16 to 17 min, (B) rapidly decreased back to 2% for re-equilibration. The column used was a Waters CORTECS C18 (4.6 × 50 mm, 2.7 μm), with the temperature controlled at 28 °C and samples maintained at 10 °C. A 15 min equilibration period was allowed between each injection to ensure consistent performance. The QDa mass spectrometer operated in negative mode, with the capillary voltage set at 0.8 kV and the cone voltage at 15 V. Mass spectra were acquired in the range of *m/z* 100–400, with selected ion recording (SIR) at *m/z* 227 for the specific quantification of *t*RSV, providing a robust and sensitive method for analyzing the presence and concentration of this compound in the wine samples [262–264].

Our study included both white (W1 and W3) and red (W2 and W4) varieties to evaluate the presence and concentration of *t*RSV. The calibration curve parameters, indicative of the method's current performance, are detailed, providing insight into the potential for precise quantification. It is important to note that while this analysis offers valuable initial data, the method is undergoing further validation to refine its accuracy and reliability. The results herein serve as a foundation for subsequent validation efforts and offer an early look at *t*RSV levels in these wines, contributing to the broader understanding of its distribution in Romanian viniculture.

The calibration curve (Figure 2) parameters presented in Table 1 demonstrate a robust analytical method for the quantification of *t*RSV. The wide linearity range of 15.259 to 244.141 ng/mL indicates that the method is capable of accurately measuring *t*RSV across a range of concentrations that are relevant for wine samples. This range comfortably exceeds the expected concentration levels in typical wine samples, allowing for precise quantification even in wines with lower *t*RSV content.

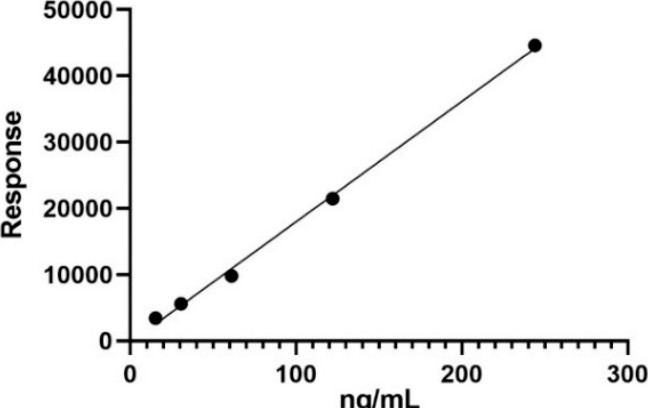

**Figure 2.** Calibration curve of *trans*-resveratrol displaying the detector response across a range of standard concentrations.

**Table 1.** Summary of calibration curve parameters for *trans*-resveratrol quantification.

| Parameter | Values |
| --- | --- |
| Linearity range [ng/mL] | 15.259–244.141 |
| $R^2$ | 0.9979 |
| Equation | $Y = 181.7X − 197.6$ |
| LOD [ng/mL] | 16.197 |
| LOQ [ng/mL] | 49.081 |

LOD: Limit of detection; LOQ: limit of quantification; $R^2$: correlation coefficient.

$R^2$ of 0.9979 reflects a high degree of linearity, suggesting that the response is directly proportional to the concentration over the range tested. This is critical for ensuring that the quantification is reliable and that the method can be used confidently for routine analysis.

The equation of the calibration curve, $Y = 181.7X − 197.6$, provides the relationship between the instrument's response and the concentration of $t$RSV. The positive slope (181.7) signifies a strong response to increasing concentrations, while the intercept (−197.6) is relatively small compared to the slope, indicating minimal background noise or interference.

LOD and LOQ are also indicators of the method's sensitivity. With a LOD of 16.197 ng/mL and a LOQ of 49.081 ng/mL, the method is sensitive enough to detect and quantify even small amounts of $t$RSV. This is particularly important for scientific and regulatory purposes where it is necessary to detect all ranges of concentrations present in various wine samples.

The UHPLC chromatogram presents a distinct peak corresponding to the pure $t$RSV reference sample. The specificity of the peak is enhanced by the MS detection of the $m/z$ 227 molecular ion, characteristic of $t$RSV. This suggests that the UHPLC–MS method provides high specificity, primarily due to the mass detection at $m/z$ 227, ensuring that the observed peak can be confidently attributed to $t$RSV without interference from other substances. Such specificity is critical for accurate quantification in complex matrices like wine (Figure 3).

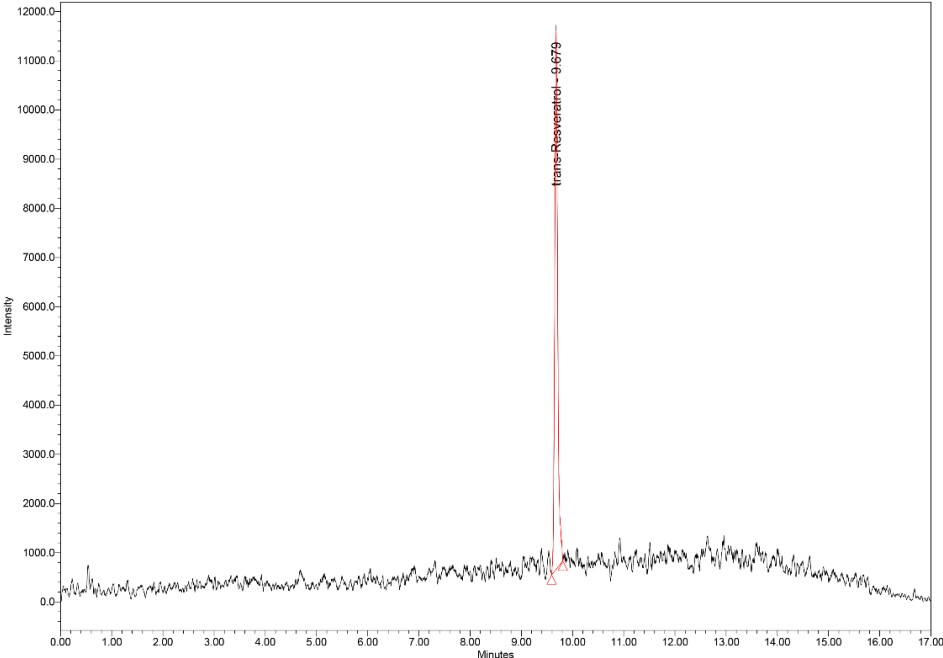

**Figure 3.** Chromatogram of a *trans*-resveratrol standard showing a prominent peak at 9.67 min, indicative of its purity and concentration in the reference sample.

In addition to the distinct peak at the specific retention time (RT), the confirmation of $t$RSV in the samples was further substantiated by comparing the mass spectrum obtained between $m/z$ 100 and 400. This range was meticulously examined to match the ionization pattern of the $t$RSV reference with the sample peaks. By ensuring that the spectra of the samples aligned with that of the reference $t$RSV, particularly the peak at $m/z$ 227, the presence of $t$RSV could be confirmed with greater certainty. This comparative approach across the broad mass range enhances the reliability of the identification, ruling out the possibility of coincidental matches or false positives that might arise from other compounds eluting at similar RTs (Figure 4).

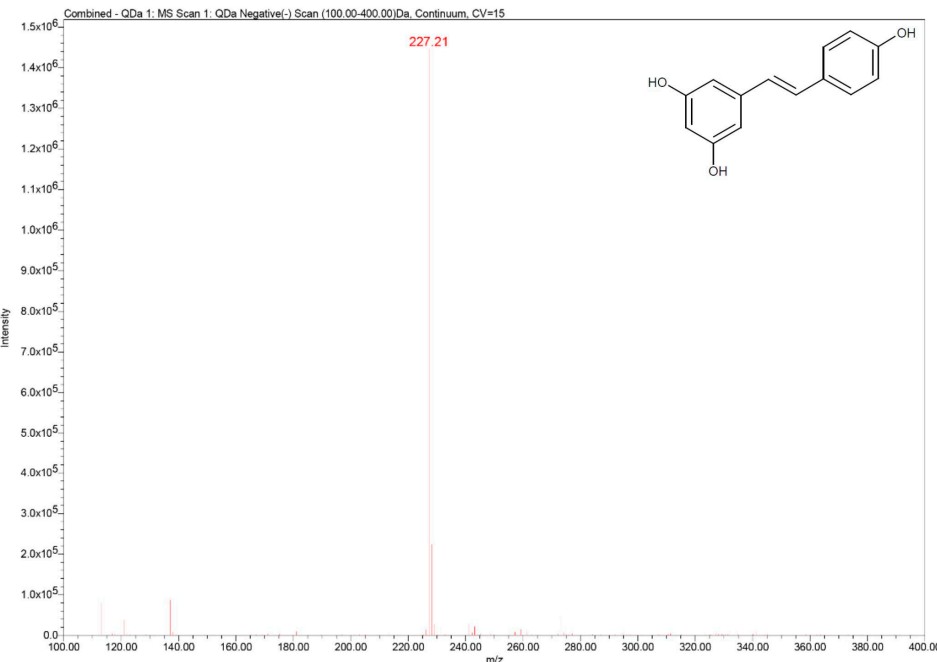

**Figure 4.** Mass spectrum of *trans*-resveratrol illustrating the quantification of molecular ion peak at *m*/z 227, used for both confirmation and quantification of the analyte.

In Figure 4, we observe the elution profiles of four different wine samples labeled W1 through W4. Each chromatogram is overlaid to facilitate comparison, particularly focusing on the *t*RSV peak. For all samples, the *t*RSV peak was identified at the same RT, which is consistent with the reference peak, suggesting successful identification of *t*RSV in each wine. The peak intensities vary among the wine samples, which may reflect the different concentrations of *t*RSV in each type of wine. The RT's consistency across all samples is a good indicator of the method's reproducibility. Furthermore, the distinct peaks observed for *t*RSV in each sample, without interference from other peaks, confirm the specificity of the method, especially considering the mass detection confirmation at *m*/z 227. This comparative display allows for a visual assessment of the presence and relative quantity of *t*RSV across the different wine samples, providing an at-a-glance understanding of how *t*RSV concentrations may differ between white wines and RWs, as well as among different varieties. The clean baselines and sharp peak shapes are indicative of a well-optimized UHPLC method, essential for accurate quantification and comparison between samples (Figure 5).

Sample W1 (Fetească Regală variety) registers the highest concentration of *t*RSV among the white wines, suggesting that this varietal may naturally accumulate more of this compound or that the vinification methods have been conducive to its retention. Moving to W3 (Dry Muscat variety), we can see that the *t*RSV content is significantly lower, which may reflect inherent varietal differences or distinct processing techniques affecting the final *t*RSV levels. When looking at the RWs, W2 (Fetească Neagră variety) contains the highest concentration of *t*RSV in the selection, aligning with the general observation that RWs often have higher levels of *t*RSV due to the extended contact with grape skins during fermentation, which typically enriches the wine with more phenolic compounds. W4 (Cabernet Sauvignon variety), however, has a notably lower level of *t*RSV, which might be influenced by a myriad of factors, including viticultural conditions, grape harvesting parameters, or the specific methodologies employed during the winemaking process (Table 2).

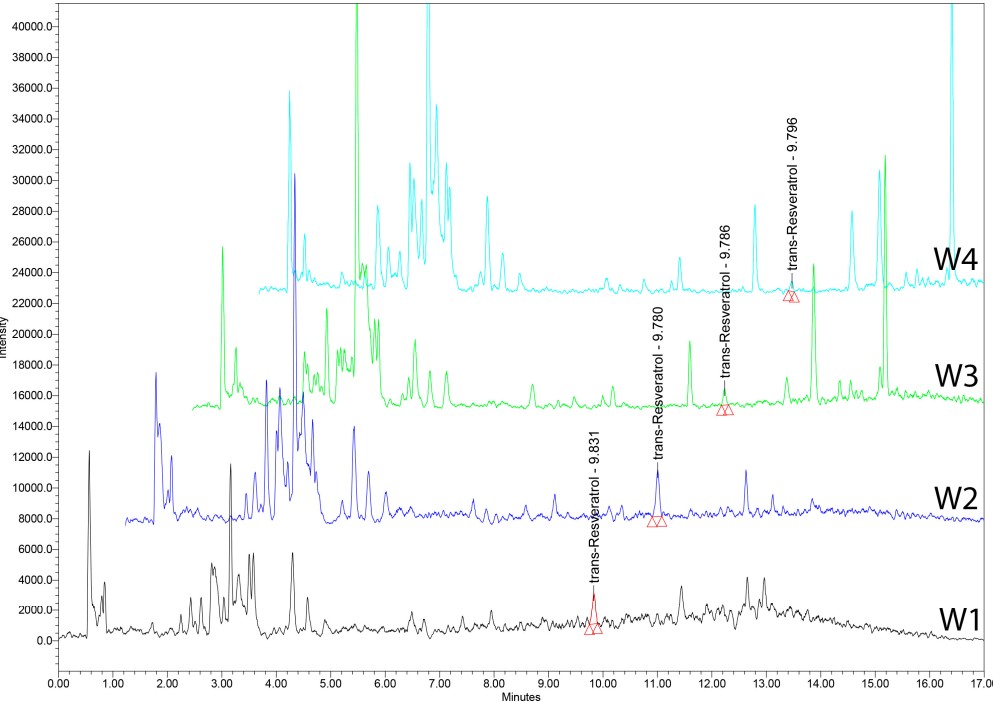

**Figure 5.** Overlay of UHPLC chromatograms for Romanian wine samples, with the *trans*-resveratrol peak identified at the specific retention time across all samples. UHPLC: Ultra-high-performance liquid chromatography; W1: Fetească Regală; W2: Fetească Neagră; W3: Dry Muscat; W4: Cabernet Sauvignon.

**Table 2.** *Trans*-resveratrol amounts detected in Romanian wine samples.

| Sample | *Trans*-Resveratrol Amount [ng/mL] |
|---|---|
| W1 | 243.291 ± 2.51 |
| W2 | 372.042 ± 2.75 |
| W3 | 108.487 ± 1.98 |
| W4 | 69.166 ± 1.05 |

White wine samples: W1: Fetească Regală; W3: Dry Muscat. Red wine samples: W2: Fetească Neagră; W4: Cabernet Sauvignon.

The low SD associated with each measurement implies that the method used for quantification is both precise and reliable. These data provide valuable insights for winemaking strategies, could inform consumer choices regarding healthful components in wine, and may stimulate additional research into the beneficial properties of wine polyphenols.

## 19. Conclusions

Stilbenoids are natural compounds of the C6–C2–C6 type, consisting of two benzene nuclei linked together by an ethylene bridge. The double bond between the two benzene nuclei can have an *E* (*trans*) or *Z* (*cis*) configuration. The *trans* configuration is the predominant form, much more pharmacologically active. For their role as phytoalexins, hormones that trigger defense mechanisms, stilbenoids are synthesized by the plant kingdom in response to external stimuli, such as infection or UV radiation. More than 1000 stilbenoids have been isolated and identified, the best known of which are the natural components RSV, piceid, piceatannol, pterostilbene, astringin, rapontigenol, viniferin, pallidol, hopeaphenol. Grapes are the main source of natural stilbenoids. RSV is a potential chemopreventive agent, both in vivo and in vitro, considering its inhibitory effects on cellular processes associated with cancer induction and progression. The influence of RSV on numerous enzyme systems explains its antioxidant, antimutagenic, and anticarcinogenic

potential. In some bioassay systems, RSV has been shown to provide protection against several types of cancer, e.g., leukemia and lymphoma, glioma, glioblastoma and neuroblastoma, breast cancer, hepatocellular carcinoma, pancreatic cancer, gastric carcinoma, colorectal cancer, lung carcinoma, cervical cancer, oral squamous cell carcinoma, prostate cancer. RSV has an antioxidant effect but also numerous pharmacological actions useful for maintaining cardiovascular health and protection against aging, especially against diseases associated with aging, neurodegeneration (AD, PD), neurotoxicity, brain injuries, inflammatory, metabolic (hyperlipidemia, atherosclerosis, obesity, hepatic steatosis, synaptic impairment, PCOS) or immune diseases, and some types of cancer. RSV modulates signaling pathways that limit the spread of tumor cells, protects nerve cells from damage, is useful in the prevention of diabetes, and generally acts as an anti-aging agent that alleviates the symptoms of aging. It was highlighted that RSV could ameliorate the consequences of an unhealthy lifestyle caused by an exaggerated caloric intake. Also, RSV exhibited bone protection, wound healing, and antimicrobial and antiviral properties. RSV has a generally non-toxic profile but requires careful dosing to maximize benefits while minimizing adverse effects. RSV bioavailability varies among individuals and is influenced by factors like diet composition and metabolism. Compared with the oral route, RSV has a good skin penetrating capacity and a low rate of degradation, ensuring an improved and long-lasting effect. The use of nanoformulations and advanced delivery systems enhances RSV's stability and tissue targeting. Moderate consumption of RW over a long period of time protects the body against CHDs and could be the main factor responsible for the so-called "French paradox". The UHPLC–MS analysis explored the influence of grape type and wine style on *t*RSV concentrations and enhanced the knowledge of how this phenolic compound varies within Romanian wine samples.

**Author Contributions:** Conceptualization, L.E.B., I.B. and G.D.M.; methodology, A.B., A.-E.S., A.R., A.D. and M.V.C.; writing—original draft preparation, L.E.B., A.B. and G.D.M.; writing—review and editing, A.B., I.B. and G.D.M.; supervision, L.E.B., I.B. and C.B.; funding acquisition, I.B. and A.-E.S. All authors have read and agreed to the published version of the manuscript.

**Funding:** This research received no external funding.

**Institutional Review Board Statement:** Not applicable.

**Informed Consent Statement:** Not applicable.

**Data Availability Statement:** Data described in the manuscript will be made publicly and freely available without restriction at: https://docs.google.com/document/d/1vVW1wFax-OrqwMIpVTSpMtgp8HrwQqx4r/edit?usp=sharing&ouid=106104952021876684289&rtpof=true&sd=true (accessed on 25 April 2024).

**Conflicts of Interest:** The authors declare no conflicts of interest.

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
