# Peer review of "Resveratrol: A Review on the Biological Activity and Applications"

_applsci, doi:10.3390/app14114534_

Round 1
Reviewer 1 Report
Comments and Suggestions for Authors
Paper entitled “Resveratrol: A Review on the Biological Activity and Applications” by Ludovic Everard Bejenaru et al. reviews the evidence supporting the beneficial effect of resveratrol in various pathological conditions (neoplastic diseases, neurodegeneration, brain injury, metabolic syndrome etc.). The paper also brings UHPLC–MS analysis of trans-resveratrol in Romanian wine samples.
Comments
• It is recommended to use another abbreviation for resveratrol (e.g., ReSV or resv.) because RSV is a common abbreviation for Respiratory Syncytial Virus
• It is recommended to change Anti-Tumoral Properties to Antitumor Properties or Anticancer Activity
• The review work is very extensive, and some parts are well-composed (especially the area of resveratrol's antitumor activity is described in detail; almost half of the work deals with it), but some important areas of its activity are very narrowly described (e.g., antioxidant, anti-inflammatory, antimicrobial, antidiabetic activity) etc.
• It is suggested that the review paper focuses on a narrower field of action (e.g., only recent research on anticancer activity or the like, as the impact of resveratrol on human health has been widely examined so far; MDPI journals alone include almost 2000 papers on resveratrol).
• Another possibility to improve the work is to limit the literature review to a very recent period (e.g., the last few years) and study different areas of resveratrol's impact on human health, including mechanisms of action.
• Many important recent works (even review papers) concerning the topic of this article are not included in the work and are not in the list of references. This should be taken into consideration.
These are, for example, the following papers:
• Koushki et al. Resveratrol: A miraculous natural compound for diseases treatment, Food Sci Nutr. 2018 Oct 26;6(8):2473-2490. doi: 10.1002/fsn3.855.
• Kaur A et al. Resveratrol: A Vital Therapeutic Agent with Multiple Health Benefits. Drug Res 2022; 72: 5–17
• Alesci, A et al. Resveratrol and Immune Cells: A Link to Improve Human Health. Molecules 2022, 27, 424. https://doi.org/10.3390/ molecules27020424
• Meng et al. Health Benefits and Molecular Mechanisms of Resveratrol: A Narrative Review. Foods 2020, 9, 340; doi:10.3390/foods9030340
• Kursvietiene et al. Anti-Cancer Properties of Resveratrol: A Focus on Its Impact on Mitochondrial Functions, Antioxidants 2023, 12, 2056. https://doi.org/10.3390/antiox12122056
• Cecerska-Heryć et al. Can Compounds of Natural Origin Be Important in Chemoprevention? Anticancer Properties of Quercetin, Resveratrol, and Curcumin—A Comprehensive Review, IJMS, Volume 25, Issue 8 10.3390/ijms25084505
... and many other papers in MDPI and other journals
• This study also includes the chromatographic analysis of trans-resveratrol in Romanian wine samples, providing an overview of its content across different types of wine. I think that such part does not fit into this review paper and should be excluded. This research can be included in another scientific paper.

Author Response
Dear Reviewer,
First of all, we would like to address you many thanks for your accurate observations and valuable comments. We used all these and improved the paper accordingly.
All changes in the revised manuscript were marked up using the “Track Changes” function.
The following changes have been made for the Manuscript (ID: applsci-3010123):
Reviewer #1 questions/comments:
Paper entitled “Resveratrol: A Review on the Biological Activity and Applications” by Ludovic Everard Bejenaru et al. reviews the evidence supporting the beneficial effect of resveratrol in various pathological conditions (neoplastic diseases, neurodegeneration, brain injury, metabolic syndrome etc.). The paper also brings UHPLC–MS analysis of trans-resveratrol in Romanian wine samples.
- It is recommended to use another abbreviation for resveratrol (e.g., ReSV or resv.) because RSV is a common abbreviation for Respiratory Syncytial Virus.
Answer:
Thank you very much for your suggestion. “RSV” abbreviation is also accepted for resveratrol. Most of the articles cited in the References list use the abbreviation “RSV” for resveratrol.
- It is recommended to change Anti-Tumoral Properties to Antitumor Properties or Anticancer Activity.
Answer:
Thank you very much for your suggestion. The terms “anti-tumoral” and “anti-tumor” have been changed to “antitumor”.
- The review work is very extensive, and some parts are well-composed (especially the area of resveratrol’s antitumor activity is described in detail; almost half of the work deals with it), but some important areas of its activity are very narrowly described (e.g., antioxidant, anti-inflammatory, antimicrobial, antidiabetic activity) etc.
Answer:
Thank you very much for your observation. The presentation on the antidiabetic, immunostimulatory, antioxidant, bone protective, wound healing, anti-inflammatory, and antimicrobial activity of RSV has been extended, citing representative references.
- It is suggested that the review paper focuses on a narrower field of action (e.g., only recent research on anticancer activity or the like, as the impact of resveratrol on human health has been widely examined so far; MDPI journals alone include almost 2000 papers on resveratrol).
Answer:
Thank you very much for your suggestion. Our paper has been conceived as an extensive review from the point of view of biological activity and applications of resveratrol. All cited references in our manuscript contribute to the scholarly content of the paper, avoid bias (self-citations, journal citations, school of thought, etc.), and reflect the current state of knowledge in the field. Considering the COPE (https://publicationethics.org) standards of publication ethics, only the most relevant citations have been kept.
- Another possibility to improve the work is to limit the literature review to a very recent period (e.g., the last few years) and study different areas of resveratrol’s impact on human health, including mechanisms of action.
Answer:
Thank you very much for your suggestion. Our paper has been conceived as an extensive review from the point of view of biological activity and applications of resveratrol. All cited references in our manuscript contribute to the scholarly content of the paper, avoid bias, and reflect the current state of knowledge in the field.
- Many important recent works (even review papers) concerning the topic of this article are not included in the work and are not in the list of references. This should be taken into consideration.
These are, for example, the following papers:
- Koushki et al. Resveratrol: A miraculous natural compound for diseases treatment, Food Sci Nutr. 2018 Oct 26;6(8):2473-2490. doi: 10.1002/fsn3.855.
- Kaur A et al. Resveratrol: A Vital Therapeutic Agent with Multiple Health Benefits. Drug Res 2022; 72: 5–17.
- Alesci, A et al. Resveratrol and Immune Cells: A Link to Improve Human Health. Molecules 2022, 27, 424. https://doi.org/10.3390/molecules27020424.
- Meng et al. Health Benefits and Molecular Mechanisms of Resveratrol: A Narrative Review. Foods 2020, 9, 340; doi:10.3390/foods9030340.
- Kursvietiene et al. Anti-Cancer Properties of Resveratrol: A Focus on Its Impact on Mitochondrial Functions, Antioxidants 2023, 12, 2056. https://doi.org/10.3390/antiox12122056.
- Cecerska-Heryć et al. Can Compounds of Natural Origin Be Important in Chemoprevention? Anticancer Properties of Quercetin, Resveratrol, and Curcumin—A Comprehensive Review, IJMS, Volume 25, Issue 8 10.3390/ijms25084505.
... and many other papers in MDPI and other journals.
Answer:
Thank you very much for your observation. Important recent works, even review papers, concerning the topic of article have been included in the work and in the list of references. (The suggested papers have also been cited in the manuscript). All cited references in our manuscript contribute to the scholarly content of the paper, avoid bias (self-citations, journal citations, school of thought, etc.), and reflect the current state of knowledge in the field. Considering the COPE (https://publicationethics.org) standards of publication ethics, only the most relevant citations have been kept.
- This study also includes the chromatographic analysis of trans-resveratrol in Romanian wine samples, providing an overview of its content across different types of wine. I think that such part does not fit into this review paper and should be excluded. This research can be included in another scientific paper.
Answer:
Thank you very much for your suggestion. Our paper has been conceived as a review on the biological activity and applications of resveratrol; the applications also include some important analytical aspects targeting the identification and quantification of resveratrol in Romanian wine samples by a modern and very useful technique (UHPLC–MS). The review covers all aspects of applied biology and applied chemistry concerning the importance of resveratrol as a natural compound.
We have also introduced other additions/modifications that we hope will improve the quality of the manuscript:
▪ All Figures have been renumbered accordingly.
▪ The Reference list has been entirely checked and renumbered accordingly.
▪ All abbreviations have been defined the first time they appear in the text.
▪ Some grammar, stylistic or spelling errors have been corrected.
Kind regards,
Ionela BELU, PhD
Corresponding Author

Reviewer 2 Report
Comments and Suggestions for Authors
The review Resveratrol: A Review on the Biological Activity and Applications of Ludovic Everard Bejenaru et al. focuses on the multiple activity of resveratrol and its health beneficial effects.
The manuscript is very interseting and useful but it needs to be improved before publishing.
There are some typos.
Above all it must be improved in content. Below are indications on the changes to be made.
At first, it is necessary to strongly motivate the validity and innovativeness of the proposed work. First, refer to other similar reviews (add references) and amplify the originality of this review study respect to the others.
The work can be accepted because it is in line with the requirements of the journal, but it is necessary to keep in mind substantial changes and minor corrections without which the manuscript is not to be considered.
Below I report some small typing and conceptual errors and some important changes that must be made before accepting the work.
MAJOR
- explain well the innovativeness of this review compared to other current ones
- explain why, given such powerful in vitro activity on many pharmacological targets, resveratrol is not a good drug. Provide some references and comment on the pharmacokinetics.
- How can draggability be improved?
- the manuscript lacks figures and tables that can improve understanding of the contents. They are only in the analytical section, but it would be very useful to insert images that immediately give an idea of the contents
- given the complexity of the work, it would also be appropriate to include an index
MINOR
- no comma in the sentence page 1, line 26
- use t in italic for tRSV and c for cis cRSV along the text (from page 1, line 34)
- page 1, line 45: use: (RSV, 3,5,4’-trihydroxystilbene)
- page 2, lines 61-64; introduce here something about the reason of “FRENCH paradox”
- from page 2, line 76: pay attention to the brackets, sometimes they are not useful. Eg, (COX), (an enzyme…), (NADH)-… Verify along the text
- page 3, line 117. Use a new paragraph and enlarge the concept
- page 3, line 130. no :
- Idem page 4, line 164
- Page 19, line 930 and 941: why do you use resveratrol and not RSV?
- Page 20, paragraph 9. The antioxidant activity of RSV is very important. Stress these contents
- Page 22, line 1099 and page 23, line 1113- use t instead of trans
- Page 27, Conclusions. They are not exhaustive of the work and the last sentence is not adequate. Please ameliorate them.
Author Response
Dear Reviewer,
First of all, we would like to address you many thanks for your accurate observations and valuable comments. We used all these and improved the paper accordingly.
All changes in the revised manuscript were marked up using the “Track Changes” function.
The following changes have been made for the Manuscript (ID: applsci-3010123):
Reviewer #2 questions/comments:
The review Resveratrol: A Review on the Biological Activity and Applications of Ludovic Everard Bejenaru et al. focuses on the multiple activity of resveratrol and its health beneficial effects.
The manuscript is very interesting and useful but it needs to be improved before publishing.
There are some typos.
Answer:
Thank you very much for your suggestion. All typographical errors have been corrected.
Above all it must be improved in content. Below are indications on the changes to be made.
At first, it is necessary to strongly motivate the validity and innovativeness of the proposed work. First, refer to other similar reviews (add references) and amplify the originality of this review study respect to the others.
Answer:
Thank you very much for your observation. Important recent works and similar review papers have been included in the manuscript and in the list of references. The originality of this review study has been amplified and documented accordingly. All cited references in our manuscript contribute to the scholarly content of the paper, avoid bias (self-citations, journal citations, school of thought, etc.), and reflect the current state of knowledge in the field. Considering the COPE (https://publicationethics.org) standards of publication ethics, only the most relevant citations have been kept.
The work can be accepted because it is in line with the requirements of the journal, but it is necessary to keep in mind substantial changes and minor corrections without which the manuscript is not to be considered.
Below I report some small typing and conceptual errors and some important changes that must be made before accepting the work.
MAJOR
- Explain well the innovativeness of this review compared to other current ones.
Answer:
Thank you very much for your suggestion. Our paper has been conceived as an exhaustive review on the biological activity and applications of resveratrol; the applications also include some important analytical aspects targeting the identification and quantification of resveratrol in Romanian wine samples by a modern and very useful technique (UHPLC–MS). The review covers all aspects of applied biology and applied chemistry concerning the importance of resveratrol as a natural compound.
- Explain why, given such powerful in vitro activity on many pharmacological targets, resveratrol is not a good drug. Provide some references and comment on the pharmacokinetics.
Answer:
A new section (“15. Risks and Interactions of Resveratrol”) has been included in the manuscript to explain why, given such powerful in vitro activity on many pharmacological targets, resveratrol is not a good drug. Comments on pharmacokinetics have been highlighted in a new section (“16. Pharmacokinetics of Resveratrol: Challenges in Absorption, Metabolism, and Therapeutic Efficacy”). Also, some references have been provided to each of the above-mentioned sections. (See pages 26 and 27, lines 1644-1727).
- How can draggability be improved?
Answer:
In response to your suggestion to include aspects on the druggability of resveratrol, we have added this information as requested in a new section “17. Strategies for Enhancing Resveratrol’s Bioavailability”. (See pages 27 and 28, lines 1728-1752).
- The manuscript lacks figures and tables that can improve understanding of the contents. They are only in the analytical section, but it would be very useful to insert images that immediately give an idea of the contents.
Answer:
Thank you very much for your suggestion. A new image (Figure 1) has been inserted in the “Introduction” section improving the understanding of the contents. (See page 2, lines 75 and 76).
- Given the complexity of the work, it would also be appropriate to include an index.
Answer:
Thank you very much for your suggestion. All abbreviations were written in accordance with the Instructions for Authors (https://www.mdpi.com/journal/applsci/instructions): “Abbreviations should be defined the first time they appear in each of three sections: the abstract; the main text; the first figure or table. When defined for the first time, the abbreviation should be added in parentheses after the written-out form.”
MINOR
- No comma in the sentence page 1, line 26.
Answer:
We made the correction. (See page 1, line 27).
- Use t in italic for tRSV and c for cis cRSV along the text (from page 1, line 34).
Answer:
Thank you very much for your observations. We made the suggested corrections throughout the entire manuscript (starting from page 1, line 34).
- Page 1, line 45: use: (RSV, 3,5,4’-trihydroxystilbene).
Answer:
We made the correction. (See page 1, line 45).
- Page 2, lines 61-64; introduce here something about the reason of “FRENCH paradox”.
Answer:
Thank you very much for your suggestion. We have briefly presented what the “French paradox” consists of. (See page 2, lines 64 and 65).
- From page 2, line 76: pay attention to the brackets, sometimes they are not useful. Eg, (COX), (an enzyme…), (NADH)-… Verify along the text.
Answer:
Thank you very much for your suggestion. We verified the manuscript and made the necessary corrections (starting from page 3, line 102).
- Page 3, line 117. Use a new paragraph and enlarge the concept.
Answer:
Thank you very much for your observation. A new paragraph has been used and the concept has been enlarged. (See page 3, lines 136-143).
- Page 3, line 130. no :
Answer:
We made the correction. (See page 4, line 165).
- Idem page 4, line 164.
Answer:
We made the correction. (See page 4, line 199).
- Page 19, line 930 and 941: why do you use resveratrol and not RSV?
Answer:
We made the correction. (See page 20, lines 1266 and 1277).
- Page 20, paragraph 9. The antioxidant activity of RSV is very important. Stress these contents.
Answer:
Thank you very much for your suggestion. The presentation on the antioxidant activity of RSV has been extended, citing representative references. (See pages 21 and 22, lines 1335-1358 and 1373-1375).
- Page 22, line 1099 and page 23, line 1113- use t instead of trans.
Answer:
We made the correction. (See pages 28, lines 1755 and 1769).
- Page 27, Conclusions. They are not exhaustive of the work and the last sentence is not adequate. Please ameliorate them.
Answer:
The conclusions have been rephrased according to your instructions (See page 33, lines 1960-1963, 1965-1967, 1971-1978, 1980-1982).
We have also introduced other additions/modifications that we hope will improve the quality of the manuscript:
▪ All Figures have been renumbered accordingly.
▪ The Reference list has been entirely checked and renumbered accordingly.
▪ All abbreviations have been defined the first time they appear in the text.
▪ Some grammar, stylistic or spelling errors have been corrected.
Kind regards,
Ionela BELU, PhD
Corresponding Author

Round 2
Reviewer 1 Report
Comments and Suggestions for Authors
Dear authors,
The manuscript entitled “Resveratrol: A Review on the Biological Activity and Applications” by Ludovic Everard Bejenaru et al. reviews the evidence supporting the beneficial effect of resveratrol in various pathological conditions. It has been revised and amended according to reviewer’s previous suggestions. The article has also been improved and expanded in the field of antidiabetic, immunostimulatory, antioxidant, bone protective, wound healing, anti-inflammatory, and antimicrobial activity, citing adequate literature references. In the corrected version, the article provides valuable information on biological activity and applications of resveratrol. In this form, it can be accepted for publication in the journal Applied Sciences as a review paper.
Reviewer 2 Report
Comments and Suggestions for Authors
Considering the appropriate modification of the text, tha manuscript could be accepted in the present form.